# WHAT LEARNING ALGORITHM IS IN-CONTEXT LEARNING? INVESTIGATIONS WITH LINEAR MODELS

**Ekin Akyürek**[1,2,a]   **Dale Schuurmans**[1]   **Jacob Andreas**[*2]   **Tengyu Ma**[*1,3,b]   **Denny Zhou**[*1]

[1]Google Research    [2]MIT CSAIL    [3] Stanford University    [*]collaborative advising

## ABSTRACT

Neural sequence models, especially transformers, exhibit a remarkable capacity for *in-context learning*. They can construct new predictors from sequences of labeled examples $(x, f(x))$ presented in the input without further parameter updates. We investigate the hypothesis that transformer-based in-context learners implement standard learning algorithms *implicitly*, by encoding smaller models in their activations, and updating these implicit models as new examples appear in the context. Using linear regression as a prototypical problem, we offer three sources of evidence for this hypothesis. First, we prove by construction that transformers can implement learning algorithms for linear models based on gradient descent and closed-form ridge regression. Second, we show that trained in-context learners closely match the predictors computed by gradient descent, ridge regression, and exact least-squares regression, transitioning between different predictors as transformer depth and dataset noise vary, and converging to Bayesian estimators for large widths and depths. Third, we present preliminary evidence that in-context learners share algorithmic features with these predictors: learners' late layers non-linearly encode weight vectors and moment matrices. These results suggest that in-context learning is understandable in algorithmic terms, and that (at least in the linear case) learners may rediscover standard estimation algorithms.

## 1 INTRODUCTION

One of the most surprising behaviors observed in large neural sequence models is **in-context learning** (ICL; Brown et al., 2020). When trained appropriately, models can map from sequences of $(x, f(x))$ pairs to accurate predictions $f(x')$ on novel inputs $x'$. This behavior occurs both in models trained on collections of few-shot learning problems (Chen et al., 2022; Min et al., 2022) and surprisingly in large language models trained on open-domain text (Brown et al., 2020; Zhang et al., 2022; Chowdhery et al., 2022). ICL requires a model to implicitly construct a map from in-context examples to a predictor without any updates to the model's parameters themselves. How can a neural network with fixed parameters to learn a new function from a new dataset on the fly?

This paper investigates the hypothesis that some instances of ICL can be understood as *implicit* implementation of known learning algorithms: in-context learners encode an implicit, context-dependent model in their hidden activations, and train this model on in-context examples in the course of computing these internal activations. As in recent investigations of empirical properties of ICL (Garg et al., 2022; Xie et al., 2022), we study the behavior of transformer-based predictors (Vaswani et al., 2017) on a restricted class of learning problems, here linear regression. Unlike in past work, our goal is not to understand *what* functions ICL can learn, but *how* it learns these functions: the specific inductive biases and algorithmic properties of transformer-based ICL.

In Section 3, we investigate theoretically what learning algorithms transformer decoders can implement. We prove by construction that they require only a modest number of layers and hidden units to train linear models: for $d$-dimensional regression problems, with $\mathcal{O}(d)$ hidden size and constant depth, a transformer can implement a single step of gradient descent; and with $\mathcal{O}(d^2)$ hidden size

---

[a]Correspondence to akyurek@mit.edu. Ekin is a student at MIT, and began this work while he was intern at Google Research. Code and reference implementations are released at this web page

[b]The work is done when Tengyu Ma works as a visiting researcher at Google Research.

and constant depth, a transformer can update a ridge regression solution to include a single new observation. Intuitively, $n$ steps of these algorithms can be implemented with $n$ times more layers.

In Section 4, we investigate empirical properties of trained in-context learners. We begin by constructing linear regression problems in which learner behavior is under-determined by training data (so different valid learning rules will give different predictions on held-out data). We show that model predictions are closely matched by existing predictors (including those studied in Section 3), and that they *transition* between different predictors as model depth and training set noise vary, behaving like Bayesian predictors at large hidden sizes and depths. Finally, in Section 5, we present preliminary experiments showing how model predictions are computed algorithmically. We show that important intermediate quantities computed by learning algorithms for linear models, including parameter vectors and moment matrices, can be decoded from in-context learners' hidden activations.

A complete characterization of which learning algorithms are (or could be) implemented by deep networks has the potential to improve both our theoretical understanding of their capabilities and limitations, and our empirical understanding of how best to train them. This paper offers first steps toward such a characterization: some in-context learning appears to involve familiar algorithms, discovered and implemented by transformers from sequence modeling tasks alone.

## 2 PRELIMINARIES

Training a machine learning model involves many decisions, including the choice of model architecture, loss function and learning rule. Since the earliest days of the field, research has sought to understand whether these modeling decisions can be automated using the tools of machine learning itself. Such "meta-learning" approaches typically treat learning as a **bi-level optimization** problem (Schmidhuber et al., 1996; Andrychowicz et al., 2016; Finn et al., 2017): they define "inner" and "outer" models and learning procedures, then train an outer model to set parameters for an inner procedure (e.g. initializer or step size) to maximize inner model performance across tasks.

Recently, a more flexible family of approaches has gained popularity. In **in-context learning** (ICL), meta-learning is reduced to ordinary supervised learning: a large sequence model (typically implemented as a transformer network) is trained to map from sequences $[x_1, f(x_1), x_2, f(x_2), ..., x_n]$ to predictions $f(x_n)$ (Brown et al., 2020; Olsson et al., 2022; Laskin et al., 2022; Kirsch & Schmidhuber, 2021). ICL does not specify an explicit inner learning procedure; instead, this procedure exists only implicitly through the parameters of the sequence model. ICL has shown impressive results on synthetic tasks and naturalistic language and vision problems (Garg et al., 2022; Min et al., 2022; Zhou et al., 2022).

Past work has characterized *what* kinds of functions ICL can learn (Garg et al., 2022; Laskin et al., 2022) and the distributional properties of pretraining that can elicit in-context learning (Xie et al., 2021; Chan et al., 2022). But *how* ICL learns these functions has remained unclear. What learning algorithms (if any) are implementable by deep network models? Which algorithms are actually discovered in the course of training? This paper takes first steps toward answering these questions, focusing on a widely used model architecture (the transformer) and an extremely well-understood class of learning problems (linear regression).

### 2.1 THE TRANSFORMER ARCHITECTURE

**Transformers** (Vaswani et al., 2017) are neural network models that map a sequence of input vectors $\boldsymbol{x} = [x_1, \ldots, x_n]$ to a sequence of output vectors $\boldsymbol{y} = [y_1, \ldots, y_n]$. Each **layer** in a transformer maps a matrix $H^{(l)}$ (interpreted as a sequence of vectors) to a sequence $H^{(l+1)}$. To do so, a transformer layer processes each column $\boldsymbol{h}_i^{(l)}$ of $H^{(l)}$ in parallel. Here, we are interested in *autoregressive* (or "decoder-only") transformer models in which each layer first computes a **self-attention**:

$$\boldsymbol{a}_i = \text{Attention}(\boldsymbol{h}_i^{(l)}; W^F, W^Q, W^K, W^V) \tag{1}$$

$$= W^F[\boldsymbol{b}_1, \ldots, \boldsymbol{b}_m] \tag{2}$$

where each $\boldsymbol{b}$ is the response of an "attention head" defined by:

$$\boldsymbol{b}_j = \text{softmax}\Big((W_j^Q \boldsymbol{h}_i)^\top (W_j^K H_{:i})\Big)(W_j^V H_{:i}) \,. \tag{3}$$

then applies a **feed-forward transformation:**

$$\boldsymbol{h}_i^{(l+1)} = \text{FF}(\boldsymbol{a}_i; W_1, W_2) \tag{4}$$

$$= W_1 \sigma(W_2 \lambda(\boldsymbol{a}_i + \boldsymbol{h}_i^{(l)})) + \boldsymbol{a}_i + \boldsymbol{h}_i^{(l)} \ . \tag{5}$$

Here $\sigma$ denotes a nonlinearity, e.g. a Gaussian error linear unit (GeLU; Hendrycks & Gimpel, 2016):

$$\sigma(x) = \frac{x}{2}\left(1 + \text{erf}\left(\frac{x}{\sqrt{2}}\right)\right), \tag{6}$$

and $\lambda$ denotes layer normalization (Ba et al., 2016):

$$\lambda(\boldsymbol{x}) = \frac{\boldsymbol{x} - \mathbb{E}[\boldsymbol{x}]}{\sqrt{\text{Var}[\boldsymbol{x}]}}, \tag{7}$$

where the expectation and variance are computed across the entries of $\boldsymbol{x}$. To map from $\boldsymbol{x}$ to $\boldsymbol{y}$, a transformer applies a sequence of such layers, each with its own parameters. We use $\theta$ to denote a model's full set of parameters (the complete collection of $W$ matrices across layers). The three main factors governing the computational capacity of a transformer are its **depth** (the number of layers), its **hidden size** (the dimension of the vectors $\boldsymbol{h}$), and the number of **heads** (denoted $m$ above).

## 2.2 TRAINING FOR IN-CONTEXT LEARNING

We study transformer models directly trained on an ICL objective. (Some past work has found that ICL also "emerges" in models trained on general text datasets; Brown et al., 2020.) To train a transformer $T$ with parameters $\theta$ to perform ICL, we first define a class of functions $\mathcal{F}$, a distribution $p(f)$ supported on $\mathcal{F}$, a distribution $p(x)$ over the domain of functions in $\mathcal{F}$, and a loss function $\mathcal{L}$. We then choose $\theta$ to optimize the auto-regressive objective, where the resulting $T_\theta$ is an **in-context learner**:

$$\arg\min_{\theta} \ \mathbb{E}_{\substack{\boldsymbol{x}_1,\ldots,\boldsymbol{x}_n \sim p(x) \\ f \sim p(f)}} \left[ \sum_{i=1}^{n} \mathcal{L}\left(f(\boldsymbol{x}_i), T_\theta\left([\boldsymbol{x}_1, f(\boldsymbol{x}_1)\ldots, \boldsymbol{x}_i]\right)\right) \right] \tag{8}$$

## 2.3 LINEAR REGRESSION

Our experiments focus on **linear regression** problems. In these problems, $\mathcal{F}$ is the space of linear functions $f(\boldsymbol{x}) = \boldsymbol{w}^\top \boldsymbol{x}$ where $\boldsymbol{w}, \boldsymbol{x} \in \mathbb{R}^d$, and the loss function is the squared error $\mathcal{L}(y, y') = (y - y')^2$. Linear regression is a model problem in machine learning and statistical estimation, with diverse algorithmic solutions. It thus offers an ideal test-bed for understanding ICL. Given a dataset with inputs $X = [\boldsymbol{x}_1, \ldots, \boldsymbol{x}_n]$ and $\boldsymbol{y} = [y_1, \ldots, y_n]$, the (regularized) linear regression objective:

$$\sum_i \mathcal{L}(\boldsymbol{w}^\top \boldsymbol{x}_i, y_i) + \lambda \|\boldsymbol{w}\|_2^2 \tag{9}$$

$$\text{is minimized by:} \quad \boldsymbol{w}^* = (X^\top X + \lambda I)^{-1} X^\top y \tag{10}$$

With $\lambda = 0$, this objective is known as **ordinary least squares regression** (OLS); with $\lambda > 0$, it is known as **ridge regression** (Hoerl & Kennard, 1970). (As discussed further in Section 4, ridge regression can also be assigned a Bayesian interpretation.) To present a linear regression problem to a transformer, we encode both $x$ and $f(x)$ as $d+1$-dimensional vectors: $\tilde{x}_i = [0, x_i]$, $\tilde{y}_i = [y_i, \boldsymbol{0}_d]$, where $\boldsymbol{0}_d$ denotes the $d$-dimensional zero vector.

## 3 WHAT LEARNING ALGORITHMS CAN A TRANSFORMER IMPLEMENT?

For a transformer-based model to solve Eq. (9) by implementing an explicit learning algorithm, that learning algorithm must be implementable via Eq. (1) and Eq. (4) with some fixed choice of transformer parameters $\theta$. In this section, we prove constructively that such parameterizations exist, giving concrete implementations of two standard learning algorithms. These proofs yield upper bounds on how many layers and hidden units suffice to implement (though not necessarily learn) each algorithm. Proofs are given in Appendices A and B.

## 3.1 PRELIMINARIES

It will be useful to first establish a few computational primitives with simple transformer implementations. Consider the following four functions from $\mathbb{R}^{H \times T} \to \mathbb{R}^{H \times T}$:

$\textbf{mov}(H; s, t, i, j, i', j')$: selects the entries of the $s^{\text{th}}$ column of $H$ between rows $i$ and $j$, and copies them into the $t^{\text{th}}$ column ($t \geq s$) of $H$ between rows $i'$ and $j'$, yielding the matrix:

$$\begin{bmatrix} | & H_{:i-1,t} & | \\ H_{:,:t} & H_{i':j',s} & H_{:,t+1:} \\ | & H_{j,t} & | \end{bmatrix} .$$

$\textbf{mul}(H; a, b, c, (i,j), (i',j'), (i'',j''))$: in *each* column $\boldsymbol{h}$ of $H$, interprets the entries between $i$ and $j$ as an $a \times b$ matrix $A_1$, and the entries between $i'$ and $j'$ as a $b \times c$ matrix $A_2$, multiplies these matrices together, and stores the result between rows $i''$ and $j''$, yielding a matrix in which each column has the form $[\boldsymbol{h}_{:i''-1}, A_1 A_2, \boldsymbol{h}_{j'':}]^\top$.

$\textbf{div}(H; (i,j), i', (i'',j''))$: in each column $\boldsymbol{h}$ of $H$, divides the entries between $i$ and $j$ by the absolute value of the entry at $i'$, and stores the result between rows $i''$ and $j''$, yielding a matrix in which every column has the form $[\boldsymbol{h}_{:i''-1}, \boldsymbol{h}_{i:j}/|\boldsymbol{h}_{i'}|, \boldsymbol{h}_{j'':}]^\top$.

$\textbf{aff}(H; (i,j), (i',j'), (i'',j''), W_1, W_2, b)$: in each column $\boldsymbol{h}$ of $H$, applies an affine transformation to the entries between $i$ and $j$ and $i'$ and $j'$, then stores the result between rows $i''$ and $j''$, yielding a matrix in which every column has the form $[\boldsymbol{h}_{:i''-1}, W_1 \boldsymbol{h}_{i:j} + W_2 \boldsymbol{h}_{i':j'} + b, \boldsymbol{h}_{j'':}]^\top$.

**Lemma 1.** *Each of* mov, mul, div *and* aff *can be implemented by a single transformer decoder layer: in Eq.* (1) *and Eq.* (4), *there exist matrices* $W^Q$, $W^K$, $W^V$, $W^F$, $W_1$ *and* $W_2$ *such that, given a matrix* $H$ *as input, the layer's output has the form of the corresponding function output above.* [1]

With these operations, we can implement building blocks of two important learning algorithms.

## 3.2 GRADIENT DESCENT

Rather than directly solving linear regression problems by evaluating Eq. (10), a standard approach to learning exploits a generic loss minimization framework, and optimizes the ridge-regression objective in Eq. (9) via gradient descent on parameters $\boldsymbol{w}$. This involves repeatedly computing updates:

$$\boldsymbol{w}' = \boldsymbol{w} - \alpha \frac{\partial}{\partial \boldsymbol{w}} \Big( \mathcal{L}(\boldsymbol{w}^\top \boldsymbol{x}_i, y_i) + \lambda \|\boldsymbol{w}\|_2^2 \Big) = \boldsymbol{w} - 2\alpha(\boldsymbol{x}\boldsymbol{w}^\top \boldsymbol{x} - y\boldsymbol{x} + \lambda\boldsymbol{w}) \tag{11}$$

for different examples $(\boldsymbol{x}_i, y_i)$, and finally predicting $\boldsymbol{w}'^\top \boldsymbol{x}_n$ on a new input $x_n$. A step of this gradient descent procedure can be implemented by a transformer:

**Theorem 1.** *A transformer can compute Eq.* (11) *(i.e. the prediction resulting from single step of gradient descent on an in-context example) with constant number of layers and* $O(d)$ *hidden space, where* $d$ *is the problem dimension of the input* $x$. *Specifically, there exist transformer parameters* $\theta$ *such that, given an input matrix of the form:*

$$H^{(0)} = \begin{bmatrix} \cdots & 0 & y_i & 0 & \cdots \\ & \boldsymbol{x}_i & 0 & \boldsymbol{x}_n & \end{bmatrix} , \tag{12}$$

*the transformer's output matrix* $H^{(L)}$ *contains an entry equal to* $\boldsymbol{w}'^\top \boldsymbol{x}_n$ *(Eq.* (11)*) at the column index where* $x_n$ *is input.*

## 3.3 CLOSED-FORM REGRESSION

Another way to solve the linear regression problem is to directly compute the closed-form solution Eq. (10). This is somewhat challenging computationally, as it requires inverting the regularized covariance matrix $X^\top X + \lambda I$. However, one can exploit the Sherman–Morrison formula (Sherman & Morrison, 1950) to reduce the inverse to a sequence of rank-one updates performed example-by-example. For any invertible square $A$,

$$\left(A + \boldsymbol{u}\boldsymbol{v}^\top\right)^{-1} = A^{-1} - \frac{A^{-1}\boldsymbol{u}\boldsymbol{v}^\top A^{-1}}{1 + \boldsymbol{v}^\top A^{-1}\boldsymbol{u}}. \tag{13}$$

---

[1]We omit the trivial size preconditions, e.g. mul: $(i - j = a * b, i' - j' = b * c, i'' - j'' = c * d)$.

Because the covariance matrix $X^\top X$ in Eq. (10) can be expressed as a sum of rank-one terms each involving a single training example $\boldsymbol{x}_i$, this can be used to construct an iterative algorithm for computing the closed-form ridge-regression solution.

**Theorem 2.** *A transformer can predict according to a single Sherman–Morrison update:*

$$\boldsymbol{w}' = \left(\lambda I + \boldsymbol{x}_i \boldsymbol{x}_i^\top\right)^{-1} \boldsymbol{x}_i y_i = \left(\frac{I}{\lambda} - \frac{\frac{I}{\lambda} \boldsymbol{x}_i \boldsymbol{x}_i^\top \frac{I}{\lambda}}{1 + \boldsymbol{x}_i^\top \frac{I}{\lambda} \boldsymbol{x}_i}\right) \boldsymbol{x}_i y_i \tag{14}$$

*with constant layers and $\mathcal{O}(d^2)$ hidden space. More precisely, there exists a set of transformer parameters $\theta$ such that, given an input matrix of the form in Eq. (12), the transformer's output matrix $H^{(L)}$ contains an entry equal to $\boldsymbol{w}'^\top x_n$ (Eq. (14)) at the column index where $x_n$ is input.*

**Discussion.** There are various existing universality results for transformers (Yun et al., 2020; Wei et al., 2021), and for neural networks more generally (Hornik et al., 1989). These generally require very high precision, very deep models, or the use of an external "tape", none of which appear to be important for in-context learning in the real world. Results in this section establish sharper upper bounds on the necessary capacity required to implement learning algorithms specifically, bringing theory closer to the range where it can explain existing empirical findings. Different theoretical constructions, in the context of meta-learning, have been shown for linear self-attention models (Schlag et al., 2021), or for other neural architectures such as recurrent neural networks (Kirsch & Schmidhuber, 2021). We emphasize that Theorem 1 and Theorem 2 each show the implementation of a single step of an iterative algorithm; these results can be straightforwardly generalized to the multi-step case by "stacking" groups of transformer layers. As described next, it is these iterative algorithms that capture the behavior of real learners.

## 4 WHAT COMPUTATION DOES AN IN-CONTEXT LEARNER PERFORM?

The previous section showed that the building blocks for two specific procedures—gradient descent on the least-squares objective and closed-form computation of its minimizer—are implementable by transformer networks. These constructions show that, in principle, fixed transformer parameterizations are expressive enough to simulate these learning algorithms. When trained on real datasets, however, in-context learners might implement other learning algorithms. In this section, we investigate the empirical properties of trained in-context learners in terms of their *behavior*. In the framework of Marr's (2010) "levels of analysis", we aim to explain ICL at the **computational** level by identifying the *kind of algorithms* to regression problems that transformer-based ICL implements.

### 4.1 BEHAVIORAL METRICS

Determining which learning algorithms best characterize ICL predictions requires first quantifying the degree to which two predictors agree. We use two metrics to do so:

**Squared prediction difference.** Given any learning algorithm $\mathcal{A}$ that maps from a set of input–output pairs $D = [\boldsymbol{x}_1, y_1, \ldots, \boldsymbol{x}_n, y_n]$ to a predictor $f(\boldsymbol{x}) = \mathcal{A}(D)(\boldsymbol{x})$, we define the squared prediction difference (SPD):

$$\mathrm{SPD}(\mathcal{A}_1, \mathcal{A}_2) = \underset{\substack{D=[\boldsymbol{x}_1,\ldots]\sim p(D) \\ \boldsymbol{x}'\sim p(\boldsymbol{x})}}{\mathbb{E}} \left(\mathcal{A}_1(D)(\boldsymbol{x}') - \mathcal{A}_2(D)(\boldsymbol{x}')\right)^2, \tag{15}$$

where $D$ is sampled as in Eq. (8). SPD measures agreement at the *output* level, regardless of the algorithm used to compute this output.

**Implicit linear weight difference.** When ground-truth predictors all belong to a known, parametric function class (as with the linear functions here), we may also investigate the extent to which different learners agree on the parameters themselves. Given an algorithm $\mathcal{A}$, we sample a *context* dataset $D$ as above, and an additional collection of unlabeled test inputs $D_{\mathcal{X}'} = \{\boldsymbol{x}'_i\}$. We then compute $\mathcal{A}$'s prediction on each $x'_i$, yielding a *predictor-specific dataset* $D_{\mathcal{A}} = \{(\boldsymbol{x}'_i, \hat{y}_i)\} = \left\{\left(\boldsymbol{x}_i, \mathcal{A}(D)(\boldsymbol{x}'_i)\right)\right\}$ encapsulating the function learned by $\mathcal{A}$. Next we compute the implied parameters:

$$\hat{\boldsymbol{w}}_{\mathcal{A}} = \arg\min_{\boldsymbol{w}} \sum_i (\hat{y}_i - \boldsymbol{w}^\top \boldsymbol{x}'_i)^2. \tag{16}$$

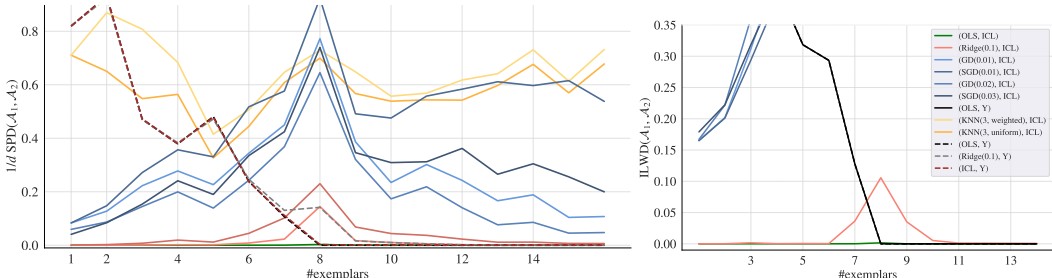

(a) Predictor–ICL fit w.r.t. prediction differences.  (b) Predictor–ICL fit w.r.t implicit weights.

Figure 1: **Fit between ICL and standard learning algorithms:** We plot (dimension normalized) SPD and ILWD values between textbook algorithms and ICL on noiseless linear regression with $d = 8$. GD($\alpha$) denotes one step of batch gradient descent and SGD($\alpha$) denotes one pass of stochastic gradient descent with learning rate $\alpha$. Ridge($\lambda$) denotes Ridge regression with regularization parameter $\lambda$. Under both evaluations, in-context learners *agree* closely with ordinary least squares, and are significantly less well approximated by other solutions to the linear regression problem.

We can then quantify agreement between two predictors $\mathcal{A}_1$ and $\mathcal{A}_2$ by computing the distance between their implied weights in expectation over datasets:

$$\text{ILWD}(\mathcal{A}_1, \mathcal{A}_2) = \mathbb{E}_D \mathbb{E}_{D_{\mathcal{X}'}} \|\hat{\boldsymbol{w}}_{\mathcal{A}_1} - \hat{\boldsymbol{w}}_{\mathcal{A}_2}\|_2^2 \ . \tag{17}$$

When the predictors are not linear, ILWD measures the difference between the closest linear predictors (in the sense of Eq. (16)) to each algorithm. For algorithms that have linear hypothesis space (e.g. Ridge regression), we will use the actual value of $\hat{\boldsymbol{w}}_{\mathcal{A}}$ instead of the estimated value.

## 4.2 EXPERIMENTAL SETUP

We train a Transformer decoder autoregresively on the objective in Eq. (8). For all experiments, we perform a hyperparameter search over depth $L \in \{1, 2, 4, 8, 12, 16\}$, hidden size $W \in \{16, 32, 64, 256, 512, 1024\}$ and heads $M \in \{1, 2, 4, 8\}$. Other hyper-parameters are noted in Appendix D. For our main experiments, we found that $L = 16, H = 512, M = 4$ minimized loss on a validation set. We follow the training guidelines in Garg et al. (2022), and trained models for $500,000$ iterations, with each in-context dataset consisting of 40 $(\boldsymbol{x}, y)$ pairs. For the main experiments we generate data according to $p(\boldsymbol{w}) = \mathcal{N}(0, I)$ and $p(\boldsymbol{x}) = \mathcal{N}(0, I)$.

## 4.3 RESULTS

**ICL matches ordinary least squares predictions on noiseless datasets.**  We begin by comparing a $(L = 16, H = 512, M = 4)$ transformer against a variety of reference predictors:

- $k$**-nearest neighbors**: In the *uniform* variant, models predict $\hat{y}_i = \frac{1}{3} \sum_j y_j$, where $j$ is the top-3 closest data point to $x_i$ where $j < i$. In the *weighted* variant, a weighted average $\hat{y}_i \propto \frac{1}{3} \sum_j |x_i - x_j|^{-2} y_j$ is calculated, normalized by the total weights of the $y_j$s.

- **One-pass stochastic gradient descent**: $\hat{y}_i = \boldsymbol{w}_i^\top x_i$ where $\boldsymbol{w}_i$ is obtained by stochastic gradient descent on the previous examples with batch-size equals to 1: $\boldsymbol{w}_i = \boldsymbol{w}_{i-1} - 2\alpha(x_{i-1}^\top \boldsymbol{w}_{i-1}^\top x_{i-1} - x_{i-1}^\top y_{i-1} + \lambda w_{i-1})$.

- **One-step batch gradient descent**: $\hat{y}_i = \boldsymbol{w}_i^\top x_i$ where $\boldsymbol{w}_i$ is obtained by one of step gradient descent on the batch of previous examples: $\boldsymbol{w}_i = \boldsymbol{w}_0 - 2\alpha(X^\top \boldsymbol{w}^\top X - X^\top Y + \lambda w_0)$.

- **Ridge regression**: We compute $\hat{y}_i = \boldsymbol{w'}^\top x_i$ where $\boldsymbol{w'}^\top = (X^\top X + \lambda I)^{-1} X^\top Y$. We denote the case of $\lambda = 0$ as **OLS**.

The agreement between the transformer-based ICL and these predictors is shown in Fig. 1. As can be seen, there are clear differences in fit to predictors: for almost any number of examples, normalized SPD and ILWD are small between the transformer and OLS predictor (with squared error less than 0.01), while other predictors (especially nearest neighbors) agree considerably less well.

When the number of examples is less than the input dimension $d = 8$, the linear regression problem is under-determined, in the sense that multiple linear models can exactly fit the in-context training

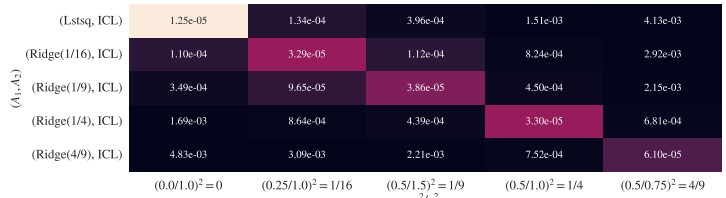

Figure 2: **ICL under uncertainty:** With problem dimension $d = 8$, and for different values of prior variance $\tau^2$ and data noise $\sigma^2$, we display (dimension-normalized) MSPD values for each predictor pair, where MSPD is the average SPD value over underdetermined region of the linear problem. Brightness is proportional with $\frac{1}{\text{MSPD}}$. ICL most closely follows the minimum-Bayes-risk Ridge regression output for all $\frac{\sigma^2}{\tau^2}$ values.

dataset. In these cases, OLS regression selects the *minimum-norm* weight vector, and (as shown in Fig. 1), the in-context learner's predictions are reliably consistent with this minimum-norm predictor. Why, when presented with an ambiguous dataset, should ICL behave like this particular predictor? One possibility is that, because the weights used to generate the training data are sampled from a Gaussian centered at zero, ICL learns to output the *minimum Bayes risk* solution when predicting under uncertainty. Building on these initial findings, our next set of experiments investigates whether ICL is behaviorally equivalent to Bayesian inference more generally.

**ICL matches the minimum Bayes risk predictor on noisy datasets.** To more closely examine the behavior of ICL algorithms under uncertainty, we add noise to the training data: now we present the in-context dataset as a sequence: $[\boldsymbol{x}_1, f(\boldsymbol{x}_1) + \epsilon_1, \ldots, \boldsymbol{x}_n, f(\boldsymbol{x}_n) + \epsilon_n]$ where each $\epsilon_i \sim \mathcal{N}(0, \sigma^2)$. Recall that ground-truth weight vectors are themselves sampled from a Gaussian distribution; together, this choice of prior and noise mean that the learner cannot be certain about the target function with any number of examples.

Standard Bayesian statistics gives that the optimal predictor for minimizing the loss in Eq. (8) is:

$$\hat{y} = \mathbb{E}[y|x, D]. \tag{18}$$

This is because, conditioned on $x$ and $D$, the scalar $\hat{y}(x, D) := \mathbb{E}[y|x, D]$ is the minimizer of the loss $\mathbb{E}[(y - \hat{y})^2|x, D]$, and thus the estimator $\hat{y}$ is the minimzier of $\mathbb{E}[(y - \hat{y})^2] = \mathbb{E}_{x,D}[\mathbb{E}[(y - \hat{y})^2|x, D]]$. For linear regression with Gaussian priors and Gaussian noise, the Bayesian estimator in Eq. (18) has a closed-form expression:

$$\boldsymbol{\hat{w}} = \left(X^\top X + \frac{\sigma^2}{\tau^2}I\right)^{-1} X^\top Y \; ; \qquad \hat{y} = \boldsymbol{\hat{w}}^\top \boldsymbol{x} \; . \tag{19}$$

Note that this predictor has the same form as the ridge predictor from Section 2.3, with the regularization parameter set to $\frac{\sigma^2}{\tau^2}$. In the presence of noisy labels, does ICL match this Bayesian predictor? We explore this by varying both the dataset noise $\sigma^2$ and the prior variance $\tau^2$ (sampling $\boldsymbol{w} \sim \mathcal{N}(0, \tau^2)$). For these experiments, the SPD values between the in-context learner and various regularized linear models is shown in Fig. 2. As predicted, as variance increases, the value of the ridge parameter that best explains ICL behavior also increases. For all values of $\sigma^2, \tau^2$, the ridge parameter that gives the best fit to the transformer behavior is also the one that minimizes Bayes risk. These experiments clarify the finding above, showing that ICL in this setting behaviorally matches minimum-Bayes-risk predictor. We also note that when the noise level $\sigma \to 0^+$, the Bayes predictor converges to the ordinary least square predictor. Therefore, the results on noiseless datasets studied in the beginning paragraph of this subsection can be viewed as corroborating the finding here in the setting with $\sigma \to 0^+$.

**ICL exhibits algorithmic phase transitions as model depth increases.** The two experiments above evaluated extremely high-capacity models in which (given findings in Section 3) computational constraints are not likely to play a role in the choice of algorithm implemented by ICL. But what about smaller models—does the size of an in-context learner play a role in determining the learning algorithm it implements? To answer this question, we run two final behavioral experiments: one in which we vary the *hidden size* (while optimizing the depth and number of heads as in Section 4.2), then vary the *depth* of the transformer (while optimizing the hidden size and number of heads). These experiments are conducted without dataset noise.

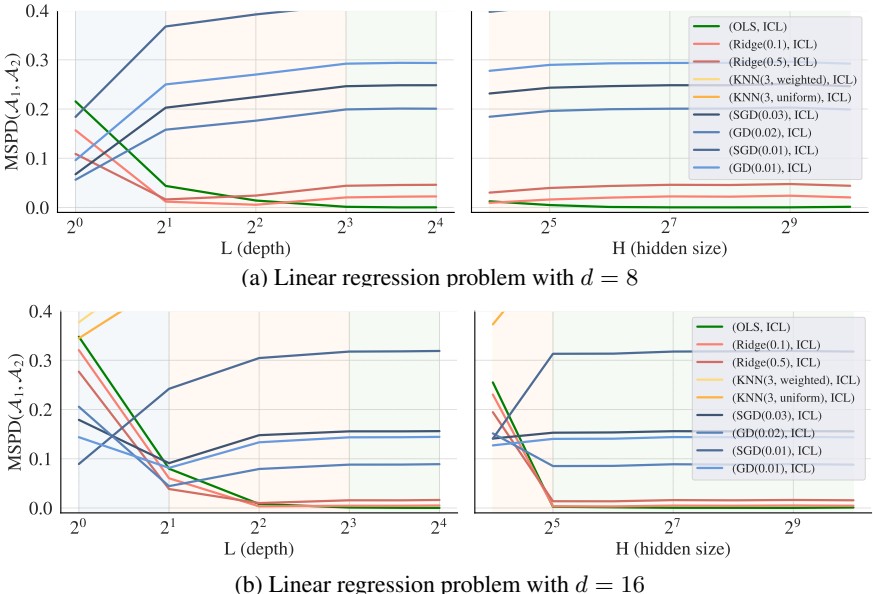

(a) Linear regression problem with $d = 8$

(b) Linear regression problem with $d = 16$

Figure 3: **Computational constraints on ICL:** We show SPD averaged over underdetermined region of the linear regression problem. In-context learners behaviorally match ordinary least squares predictors if there is enough number of layers and hidden sizes. When varying model depth (left background), algorithmic "phases" emerge: models transition between being closer to gradient descent, (red background), ridge regression (green background), and OLS regression (blue).

Results are shown in Fig. 3. When we vary the depth, learners occupy three distinct regimes: very shallow models (1L) are best approximated by a single step of gradient descent (though not well-approximated in an absolute sense). Slightly deeper models (2L-4L) are best approximated by ridge regression, while the deepest (+8L) models match OLS as observed in Fig. 3. Similar phase shift occurs when we vary hidden size in a 16D problem. Interestingly, we can read hidden size requirements to be close to ridge-regression-like solutions as $H \geq 16$ and $H \geq 32$ for 8D and 16D problems respectively, suggesting that ICL discovers more efficient ways to use available hidden state than our theoretical constructions requiring $\mathcal{O}(d^2)$. Together, these results show that ICL *does not necessarily* involve minimum-risk prediction. However, even in models too computationally constrained to perform Bayesian inference, alternative interpretable computations can emerge.

## 5 DOES ICL ENCODE MEANINGFUL INTERMEDIATE QUANTITIES?

Section 4 showed that transformers are a good fit to standard learning algorithms (including those constructed in Section 3) at the *computational* level. But these experiments leave open the question of how these computations are implemented at the **algorithmic** level. How do transformers arrive at the solutions in Section 4, and what quantities do they compute along the way? Research on extracting precise algorithmic descriptions of learned models is still in its infancy (Cammarata et al., 2020; Mu & Andreas, 2020). However, we can gain insight into ICL by inspecting learners' intermediate states: asking *what* information is encoded in these states, and *where*.

To do so, we identify two intermediate quantities that we expect to be computed by gradient descent and ridge-regression variants: the **moment vector** $X^\top Y$ and the (min-norm) least-square estimated **weight vector** $w_{\text{OLS}}$, each calculated after feeding $n$ exemplars. We take a trained in-context learner, freeze its weights, then train an auxiliary **probing** model (Alain & Bengio, 2016) to attempt to recover the target quantities from the learner's hidden representations. Specifically, the probe model takes hidden states at a layer $H^{(l)}$ as input, then outputs the prediction for target variable. We define a probe with **position-attention** that computes (Appendix E):

$$\boldsymbol{\alpha} = \text{softmax}(\boldsymbol{s}_v) \tag{20}$$

$$\hat{\boldsymbol{v}} = \text{FF}_v(\boldsymbol{\alpha}^\top W_v H^{(l)}) \tag{21}$$

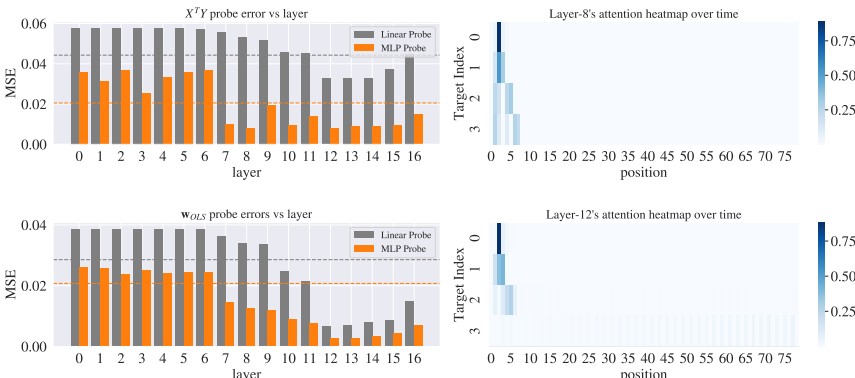

Figure 4: **Probing results on** $d = 4$ **problem:** Both moments $X^\top Y$ (top) and least-square solution $\boldsymbol{w}_{\text{OLS}}$ (middle) are recoverable from learner representations. Plots in the left column show the accuracy of the probe for each target in different model layers. Dashed lines show the best probe accuracies obtained on a *control task* featuring a fixed weight vector $w = \mathbf{1}$. Plots in the right column show the attention heatmap for the best layer's probe, with the number of input examples on the x-axis. The value of the target after $n$ exemplars is decoded primarily from the representation of $y_n$, or, after $n = d$ examplars, uniformly from $y_{n \geq 4}$.

We train this probe to minimize the squared error between the predictions and targets $\boldsymbol{v}$: $\mathcal{L}(\boldsymbol{v}, \hat{\boldsymbol{v}}) = |\boldsymbol{v} - \hat{\boldsymbol{v}}|^2$. The probe performs two functions simultaneously: its prediction error on held-out representations determines the *extent to which the target quantity is encoded*, while its attention mask, $\boldsymbol{\alpha}$ identifies the *location in which the target quantity is encoded*. For the FF term, we can insert the function approximator of our choosing; by changing this term we can determine the *manner in which the target quantity is encoded*—e.g. if FF is a linear model and the probe achieves low error, then we may infer that the target is encoded linearly.

For each target, we train a separate probe for the value of the target on *each prefix of the dataset*: i.e. one probe to decode the value of $\boldsymbol{w}$ computed from a single training example, a second probe to decode the value for two examples, etc. Results are shown in Fig. 4. For both targets, a 2-layer MLP probe outperforms a linear probe, meaning that these targets are encoded nonlinearly (unlike the constructions in Section 3). However, probing also reveals similarities. Both targets are decoded accurately deep in the network (but inaccurately in the input layer, indicating that probe success is non-trivial.) Probes attend to the correct timestamps when decoding them. As in both constructions, $X^\top Y$ appears to be computed first, becoming predictable by the probe relatively early in the computation (layer 7); while $\boldsymbol{w}$ becomes predictable later (around layer 12). For comparison, we additionally report results on a *control task* in which the transformer predicts $y$s generated with a fixed weight vector $w = \mathbf{1}$ (so no ICL is required). Probes applied to these models perform significantly worse at recovering moment matrices (see Appendix E for details).

## 6 CONCLUSION

We have presented a set of experiments characterizing the computations underlying in-context learning of linear functions in transformer sequence models. We showed that these models are capable in theory of implementing multiple linear regression algorithms, that they empirically implement this range of algorithms (transitioning between algorithms depending on model capacity and dataset noise), and finally that they can be probed for intermediate quantities computed by these algorithms.

While our experiments have focused on the linear case, they can be extended to many learning problems over richer function classes—e.g. to a network whose initial layers perform a non-linear feature computation. Even more generally, the experimental methodology here could be applied to larger-scale examples of ICL, especially language models, to determine whether their behaviors are also described by interpretable learning algorithms. While much work remains to be done, our results offer initial evidence that the apparently mysterious phenomenon of in-context learning can be understood with the standard ML toolkit, and that the solutions to learning problems discovered by machine learning researchers may be discovered by gradient descent as well.

## ACKNOWLEDGEMENTS

We thank Evan Hernandez, Andrew Drozdov, Ed Chi for their feedback on the early drafts of this paper. At MIT, Ekin Akyürek is supported by an MIT-Amazon ScienceHub fellowship and by the MIT-IBM Watson AI Lab.

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

## A  Theorem 1

The operations for 1-step SGD with single exemplar can be expressed as following chain (please see proofs for the Transformer implementation of these operations (Lemma 1) in Appendix C):

- $\text{mov}(; 1, 0, (1, 1 + d), (1, 1 + d))$      (move $\boldsymbol{x}$)
- $\text{aff}(; (1, 1 + d), (), (1 + d, 2 + d), W_1 = \boldsymbol{w})$      $(\boldsymbol{w}^\top \boldsymbol{x})$
- $\text{aff}(; (1 + d, 2 + d), (0, 1), (2 + d, 3 + d), W_1 = I, W_2 = -I)$      $(\boldsymbol{w}^\top \boldsymbol{x} - y)$
- $\text{mul}(; d, 1, 1, (1, 1 + d), (2 + d, 3 + d), (3 + d, 3 + 2d))$      $(\boldsymbol{x}(\boldsymbol{w}^\top \boldsymbol{x} - y))$
- $\text{aff}(; (), (), (3 + 2d, 3 + 3d), b = \boldsymbol{w}, )$      (write $\boldsymbol{w}$)
- $\text{aff}(; (3 + d, 3 + 2d), (3 + 2d, 3 + 3d), (3 + 3d, 3 + 4d), W_1 = I, W_2 = -\lambda)$      $(\boldsymbol{x}(\boldsymbol{w}^\top \boldsymbol{x} - y) - \lambda \boldsymbol{w})$
- $\text{aff}(; (3 + 2d, 3 + 3d), (3 + 3d, 3 + 4d), (3 + 2d, 3 + 3d), W_1 = I, W_2 = -2\alpha, )$      $(\boldsymbol{w}')$
- $\text{mov}(; 2, 1, (3 + 2d, 3 + 3d), (3 + 2d, 3 + 3d))$      (move $\boldsymbol{w}'$)
- $\text{mul}(; 1, d, 1, (3 + 2d, 3 + 3d), (1, 1 + d), (3 + 3d, 4 + 3d))$      $(\boldsymbol{w}'^\top x_2)$

This will map:

$$
\begin{bmatrix} 0 & y_1 & 0 \\ x_1 & 0 & x_2 \\ & & \\ & & \\ & & \\ & & \\ & & \\ & & \\ & & \\ & & \end{bmatrix} \mapsto \begin{bmatrix} 0 & y_1 & 0 \\ x_1 & x_1 & x_2 \\ w^\top x_1 & w^\top x_1 & w^\top x_2 \\ w^\top x_1 & w^\top x_1 - y & w^\top x_2 \\ x_1 w^\top x_1 & x_1(w^\top x_1 - y) & x_2 w^\top x_1 \\ w & w & w \\ x_1 w^\top x_1 - \lambda w & x_1(w^\top x_1 - y) - \lambda w & x_2 w^\top x_1 - \lambda w \\ w - 2\alpha(x_1 w^\top x_1 - \lambda w) & w' & w - 2\alpha(x_2 w^\top x_1 - \lambda w) \\ w - 2\alpha(x_1 w^\top x_1 - \lambda w) & w' & w' \\ (w - 2\alpha(x_1 w^\top x_1 - \lambda w))^\top x1 & w'^\top x_1 & \mathbf{w}'^\top \mathbf{x_2} \end{bmatrix}
$$

We can verify the chain of operator step-by-step. In each step, we show only the non-zero rows.

- $\text{mov}(; 1, 0, (1, 1 + d), (1, 1 + d))$      (move $\boldsymbol{x}$)

$$
\begin{bmatrix} 0 & y_1 & 0 \\ x_1 & 0 & x_2 \end{bmatrix} \mapsto \begin{bmatrix} 0 & y_1 & 0 \\ x_1 & x_1 & x_2 \end{bmatrix}
$$

- $\text{aff}(; (1, 1 + d), (), (1 + d, 2 + d), W_1 = \boldsymbol{w})$      $(\boldsymbol{w}^\top \boldsymbol{x})$

$$
\begin{bmatrix} 0 & y_1 & 0 \\ x_1 & x_1 & x_2 \end{bmatrix} \mapsto \begin{bmatrix} 0 & y_1 & 0 \\ x_1 & x_1 & x_2 \\ w^\top x_1 & w^\top x_1 & w^\top x_2 \end{bmatrix}
$$

- $\text{aff}(; (1 + d, 2 + d), (0, 1), (2 + d, 3 + d), W_1 = I, W_2 = -I)$      $(\boldsymbol{w}^\top \boldsymbol{x} - y)$

$$
\begin{bmatrix} 0 & y_1 & 0 \\ x_1 & x_1 & x_2 \\ w^\top x_1 & w^\top x_1 & w^\top x_2 \end{bmatrix} \mapsto \begin{bmatrix} 0 & y_1 & 0 \\ x_1 & x_1 & x_2 \\ w^\top x_1 & w^\top x_1 & w^\top x_2 \\ w^\top x_1 & w^\top x_1 - y_1 & w^\top x_2 \end{bmatrix}
$$

- $\text{mul}(; d, 1, 1, (1, 1 + d), (2 + d, 3 + d), (3 + d, 3 + 2d))$      $(\boldsymbol{x}(\boldsymbol{w}^\top \boldsymbol{x} - y))$

$$
\begin{bmatrix} 0 & y_1 & 0 \\ x_1 & x_1 & x_2 \\ w^\top x_1 & w^\top x_1 & w^\top x_2 \\ w^\top x_1 & w^\top x_1 - y_1 & w^\top x_2 \end{bmatrix} \mapsto \begin{bmatrix} 0 & y_1 & 0 \\ x_1 & x_1 & x_2 \\ w^\top x_1 & w^\top x_1 & w^\top x_2 \\ w^\top x_1 & w^\top x_1 - y & w^\top x_2 \\ x_1 w^\top x_1 & x_1(w^\top x_1 - y) & x_2 w^\top x_1 \end{bmatrix}
$$

- $\text{aff}(; (), (), (3 + 2d, 3 + 3d), b = \boldsymbol{w}, )$      (write $\boldsymbol{w}$)

$$
\begin{bmatrix}
0 & y_1 & 0 \\
x_1 & x_1 & x_2 \\
w^\top x_1 & w^\top x_1 & w^\top x_2 \\
w^\top x_1 & w^\top x_1 - y & w^\top x_2 \\
x_1 w^\top x_1 & x_1(w^\top x_1 - y) & x_2 w^\top x_1
\end{bmatrix}
\mapsto
\begin{bmatrix}
0 & y_1 & 0 \\
x_1 & x_1 & x_2 \\
w^\top x_1 & w^\top x_1 & w^\top x_2 \\
w^\top x_1 & w^\top x_1 - y & w^\top x_2 \\
x_1 w^\top x_1 & x_1(w^\top x_1 - y) & x_2 w^\top x_1 \\
w & w & w
\end{bmatrix}
$$

- $\texttt{aff}(; (3+d, 3+2d), (3+2d, 3+3d), (3+3d, 3+4d), W_1 = I, W_2 = -2\lambda)$ $\quad (\boldsymbol{x}(\boldsymbol{w}^\top \boldsymbol{x} - y) - 2\lambda \boldsymbol{w})$

$$
\begin{bmatrix}
0 & y_1 & 0 \\
x_1 & x_1 & x_2 \\
w^\top x_1 & w^\top x_1 & w^\top x_2 \\
w^\top x_1 & w^\top x_1 - y & w^\top x_2 \\
x_1 w^\top x_1 & x_1(w^\top x_1 - y) & x_2 w^\top x_1 \\
w & w & w
\end{bmatrix}
\mapsto
\begin{bmatrix}
0 & y_1 & 0 \\
x_1 & x_1 & x_2 \\
w^\top x_1 & w^\top x_1 & w^\top x_2 \\
w^\top x_1 & w^\top x_1 - y & w^\top x_2 \\
x_1 w^\top x_1 & x_1(w^\top x_1 - y) & x_2 w^\top x_1 \\
w & w & w \\
x_1 w^\top x_1 - \lambda w & x_1(w^\top x_1 - y) - \lambda w & x_2 w^\top x_1 - \lambda w
\end{bmatrix}
$$

- $\texttt{aff}(; (3 + 2d, 3 + 3d), (3 + 3d, 3 + 4d), (3 + 2d, 3 + 3d), W_1 = I, W_2 = -2\alpha, )$ $\qquad \color{blue}(\boldsymbol{w}')$

$$
\begin{bmatrix}
0 & y_1 & 0 \\
x_1 & x_1 & x_2 \\
w^\top x_1 & w^\top x_1 & w^\top x_2 \\
w^\top x_1 & w^\top x_1 - y & w^\top x_2 \\
x_1 w^\top x_1 & x_1(w^\top x_1 - y) & x_2 w^\top x_1 \\
w & w & w \\
x_1 w^\top x_1 - \lambda w & x_1(w^\top x_1 - y) - \lambda w & x_2 w^\top x_1 - \lambda w
\end{bmatrix}
\mapsto
$$

$$
\begin{bmatrix}
0 & y_1 & 0 \\
x_1 & x_1 & x_2 \\
w^\top x_1 & w^\top x_1 & w^\top x_2 \\
w^\top x_1 & w^\top x_1 - y & w^\top x_2 \\
x_1 w^\top x_1 & x_1(w^\top x_1 - y) & x_2 w^\top x_1 \\
w - 2\alpha(x_1 w^\top x_1 - \lambda w) & w' & w - 2\alpha(x_2 w^\top x_1 - \lambda w) \\
x_1 w^\top x_1 - \lambda w & x_1(w^\top x_1 - y) - \lambda w & x_2 w^\top x_1 - \lambda w
\end{bmatrix}
$$

- $\texttt{mov}(; 2, 1, (3 + 2d, 3 + 3d), (3 + 2d, 3 + 3d))$ $\qquad\qquad (\text{move } \boldsymbol{w}')$

$$
\begin{bmatrix}
0 & y_1 & 0 \\
x_1 & x_1 & x_2 \\
w^\top x_1 & w^\top x_1 & w^\top x_2 \\
w^\top x_1 & w^\top x_1 - y & w^\top x_2 \\
x_1 w^\top x_1 & x_1(w^\top x_1 - y) & x_2 w^\top x_1 \\
w & w & w \\
w - 2\alpha(x_1 w^\top x_1 - \lambda w) & w' & w - 2\alpha(x_2 w^\top x_1 - \lambda w) \\
x_1 w^\top x_1 - \lambda w & x_1(w^\top x_1 - y) - \lambda w & x_2 w^\top x_1 - \lambda w
\end{bmatrix}
\mapsto
$$

$$
\begin{bmatrix}
0 & y_1 & 0 \\
x_1 & x_1 & x_2 \\
w^\top x_1 & w^\top x_1 & w^\top x_2 \\
w^\top x_1 & w^\top x_1 - y & w^\top x_2 \\
x_1 w^\top x_1 & x_1(w^\top x_1 - y) & x_2 w^\top x_1 \\
w - 2\alpha(x_1 w^\top x_1 - \lambda w) & w' & w - 2\alpha(x_2 w^\top x_1 - \lambda w) \\
x_1 w^\top x_1 - \lambda w & x_1(w^\top x_1 - y) - \lambda w & x_2 w^\top x_1 - \lambda w \\
w - 2\alpha(x_1 w^\top x_1 - \lambda w) & w' & w'
\end{bmatrix}
$$

- $\texttt{mul}(; 1, d, 1, (3 + 2d, 3 + 3d), (1, 1 + d), (3 + 3d, 4 + 3d))$ $\qquad\qquad ({\boldsymbol{w}'}^\top x_2)$

$$\begin{bmatrix} 0 & y_1 & 0 \\ x_1 & x_1 & x_2 \\ w^\top x_1 & w^\top x_1 & w^\top x_2 \\ w^\top x_1 & w^\top x_1 - y & w^\top x_2 \\ x_1 w^\top x_1 & x_1(w^\top x_1 - y) & x_2 w^\top x_1 \\ w - 2\alpha(x_1 w^\top x_1 - \lambda w) & w' & w - 2\alpha(x_2 w^\top x_1 - \lambda w) \\ x_1 w^\top x_1 - \lambda w & x_1(w^\top x_1 - y) - \lambda w & x_2 w^\top x_1 - \lambda w \\ w - 2\alpha(x_1 w^\top x_1 - \lambda w) & w' & w' \end{bmatrix} \mapsto$$

$$\begin{bmatrix} 0 & y_1 & 0 \\ x_1 & x_1 & x_2 \\ w^\top x_1 & w^\top x_1 & w^\top x_2 \\ w^\top x_1 & w^\top x_1 - y & w^\top x_2 \\ x_1 w^\top x_1 & x_1(w^\top x_1 - y) & x_2 w^\top x_1 \\ w - 2\alpha(x_1 w^\top x_1 - \lambda w) & w' & w - 2\alpha(x_2 w^\top x_1 - \lambda w) \\ x_1 w^\top x_1 - \lambda w & x_1(w^\top x_1 - y) - \lambda w & x_2 w^\top x_1 - \lambda w \\ w - 2\alpha(x_1 w^\top x_1 - \lambda w) & w' & w' \\ (w - 2\alpha(x_1 w^\top x_1 - \lambda w))^\top x1 & {w'}^\top x_1 & \mathbf{w'}^\top \mathbf{x_2} \end{bmatrix}$$

We obtain the updated prediction in the last hidden unit of the third time-step. $\quad\square$

**Generalizing to multiple steps of SGD.** Since $w'$ is written in the hidden states, we may repeat this iteration to obtain $\hat{y}_3 = {w''}^\top x_3$ where $w''$ is the one step update $w' - 2\alpha(x_2 {w'}^\top x_2 - y_2 x_2 + \lambda w)$, requiring a total of $\mathcal{O}(n)$ layers for a single pass through the dataset where $n$ is the number of examplers.

As an empirical demonstration of this procedure, the accompanying code release contains a reference implementation of SGD defined in terms of the base primitive provided in an anymous links `https://icl1.s3.us-east-2.amazonaws.com/theory/{primitives,sgd,ridge}.py` (to preserve the anonymity we did not provide the library dependencies). This implementation predicts $\hat{y}_n = w_n^\top x_n$, where $w_n$ is the weight vector resulting from $n - 1$ consecutive SGD updates on previous examples. It can be verified there that the procedure requires $\mathcal{O}(n + d)$ hidden space. Note that, it is not $\mathcal{O}(nd)$ because we can reuse spaces for the next iteration for the intermediate variables, an example of this performed in $(w')$ step above highlighted with blue color.

## B    THEOREM 2

We provide a similar construction to Theorem 1 (please see proofs for the Transformer implementation of these operations in Appendix C, specifically for `div` see Appendix C.6)

- `mov(; 1, 0, (1, 1 + d), (1, 1 + d))` (move $x_1$)
- `mul(; d, 1, 1, (1, 1 + d), (0, 1), (1 + d, 1 + 2d))` $(x_1 y)$
- `aff(; (), (), (1 + 2d, 1 + 2d + d^2), b = \frac{I}{\lambda})` $(A_0^{-1} = \frac{I}{\lambda})$
- `mul(; d, d, 1, (1 + 2d, 1 + 2d + d^2), (1, 1 + d), (1 + 2d + d^2, 1 + 3d + d^2))` $(A_0^{-1} u = \frac{I}{\lambda} x_1)$
- `mul(; 1, d, d, (1, 1 + d), (1 + 2d, 1 + 2d + d^2), (1 + 3d + d^2, 1 + 4d + d^2))` $(v A_0^{-1} = x_1^\top \frac{I}{\lambda})$
- `mul(; d, 1, d, (1 + 2d + d^2, 1 + 3d + d^2), (1 + 3d + d^2, 1 + 4d + d^2), (1 + 4d + d^2, 1 + 4d + 2d^2))` $(A_0^{-1} u v A_0^{-1} = \frac{I}{\lambda} x_1 x_1^\top \frac{I}{\lambda})$
- `mul(; 1, d, 1, (1 + 3d + d^2, 1 + 4d + d^2), (1, 1 + d), (1 + 4d + 2d^2, 2 + 4d + 2d^2))` $(v^\top A_0^{-1} u = x_1^\top \frac{I}{\lambda} x_1)$
- `aff(; (1 + 4d + 2d^2, 2 + 4d + 2d^2), (), (1 + 4d + 2d^2, 2 + 4d + 2d^2), W_1 = 1, b = 1,)` $(1 + v^\top A_0^{-1} u = 1 + x_1^\top \frac{I}{\lambda} x_1)$
- `div(; (1 + 4d + d^2, 1 + 4d + 2d^2), 1 + 4d + 2d^2, (2 + 4d + 2d^2, 2 + 4d + 3d^2))` (right term)
- `aff(; (1 + 2d, 1 + 2d + d^2), (2 + 4d + 2d^2, 2 + 4d + 3d^2), (1 + 2d, 1 + 2d + d^2), W_1 = I, W_2 = -I)` $(A_1^{-1})$
- `mul(; d, d, 1, (1 + 2d, 1 + 2d + d^2), (1, 1 + d), (2 + 4d + 3d^2, 2 + 5d + 3d^2))` $(A_1^{-1} x_1)$
- `mul(; d, 1, 1, (2 + 4d + 3d^2, 2 + 5d + 3d^2), (0, 1), (2 + 4d + 3d^2, 2 + 5d + 3d^2))` $(A_1^{-1} x_1 y_1)$

- $\mathtt{mov}(; 2, 1, (2 + 4d + 3d^2, 2 + 5d + 3d^2), (2 + 4d + 3d^2, 2 + 5d + 3d^2))$       (move $\boldsymbol{w}'$)

- $\mathtt{mul}(; d, 1, 1(2 + 4d + 3d^2, 2 + 5d + 3d^2), (1, 1 + d), (2 + 5d + 3d^2, 3 + 5d + 3d^2))$    $(\boldsymbol{w}'^{\top} x_2)$

Note that, in contrast to Appendix A, we need $\mathcal{O}(d^2)$ space to implement matrix multiplications. Therefore over-all required hidden size is $\mathcal{O}(d^2)$      $\square$

As Theorem 1, generalizing it to multiple iterations will at least require $\mathcal{O}(n)$ layers, as we repeat the process for the next examplar.

## C    LEMMA 1

All of the operators mentioned in this lemma share a common computational structure, and can in fact be implemented as special cases of a "base primitive" we call RAW (for Read-Arithmetic-Write). This operator may also be useful for future work aimed at implementing other algorithms.

The structure of our proof of Lemma 1 is as follows:

1. Motivation of the base primitive RAW.
2. Formal definition of RAW.
3. Definition of dot, aff, mov in terms of RAW.
4. Implementation of RAW in terms of transformer parameters.
5. Brief discussion of how to parallelize RAW, making it possible to implement mul.
6. Seperate proof for div by utilizing layer norm.

### C.1    RAW OPERATOR: INTUITION

At a high level, all of the primitives in Lemma 1 involve a similar sequence of operations:

**1) Operators read some hidden units from the current or previous timestep:** dot and aff read from two subsets of indices in the current hidden state $\boldsymbol{h}_t$[2], while mov reads from a previous hidden state $\boldsymbol{h}_{t'}$. This selection is straightforwardly implemented using the attention component of a transformer layer.

We may notate this reading operation as follows:

$$\underbrace{\left( \frac{1}{W_a} \sum_{k \in K(i)} \boldsymbol{h}_k^{(l)}[\mathsf{r}] \right)}_{\text{Read with Attention}} . \tag{22}$$

Here $\mathsf{r}$ denotes a list of indice to read from, and $K$ denotes a map from *current timesteps* to *target timesteps*. For convenience, we use Numpy-like notation to denote indexing into a vector with another vector:

**Definition C.1** (Bracket). $\boldsymbol{x}[.]$ is Python index notation where the resulting vector, $\boldsymbol{x}' = \boldsymbol{x}[\mathsf{r}]$:

$$\boldsymbol{y}_j = \boldsymbol{x}_{r_j} \quad j = 1, .... |\mathsf{r}|$$

The first step of our proof below shows that the attention output $\boldsymbol{a}^{(l)}$ can compute the expression above.

**2) Operators perform element-wise arithmetic between the quantity read in step 1 and another set of entries from the current timestep:** This step takes different forms for aff and mul (mov ignores values at the current timestep altogether).

---

[2]For notational convenience, we will use $\boldsymbol{h}$ to refer to sequence of hidden states (instead of $H$ in Eq. (1).), $\boldsymbol{h}_{t'}$ will be the hidden state at time step $t'$

$$\underbrace{\left(\frac{W_a}{|K(i)|}\sum_{k\in K(i)}\boldsymbol{h}_k^{(l)}[\mathsf{r}]\right)}_{\text{Read with Attention}}\odot\ W\boldsymbol{h}_i^{(l)}[\mathsf{s}]\quad\text{(multiplicative form)}\tag{23}$$

$$\underbrace{\left(\frac{W_a}{|K(i)|}\sum_{k\in K(i)}\boldsymbol{h}_k^{(l)}[\mathsf{r}]\right)}_{\text{Read with Attention}}+\ W\boldsymbol{h}_i^{(l)}[\mathsf{s}]\quad\text{(additive form)}\tag{24}$$

The second step of the proof below computes these operations inside the MLP component of the transformer layer.

**3) Operators reduce, then write to the current hidden state**  Once the underlying element-wise operation calculated, the operator needs to **write** these values to the some indices in current hidden state, defined by a list of indices w. Writing might be preceded by a **reduction** state (e.g. for computing dot products), which can be expressed generically as a linear operator $W_o$. The final form of the computation is thus:

$$\boldsymbol{h}_i^{(l+1)}[\mathsf{w}]\leftarrow\left(W_o\underbrace{\overbrace{\left(\frac{W_a}{|K(i)|}\sum_{k\in K(i)}\boldsymbol{h}_k^{(l)}[\mathsf{r}]\right)}_{\text{Read with Attention}}\circledast W\boldsymbol{h}_i^{(l)}[\mathsf{s}]}^{\text{Elementwise operation}}\right)\tag{25}$$

Here, $\leftarrow$ means that the other indices $i\notin w$ are copied from $h^{l-1}$.

## C.2  RAW OPERATOR DEFINITION

We denote this "master operator" as RAW:

**Definition C.2.** $\mathsf{RAW}(\boldsymbol{h};\circledast,\mathsf{s},\mathsf{r},\mathsf{w},W_o,W_a,W,K)$ is a function $\mathbb{R}^{H\times T}\mapsto\mathbb{R}^{H\times T}$. It is parameterized by an elementwise operator $\circledast\in\{+,\odot\}$, three matrices $W\in\mathbb{R}^{d\times|\mathsf{s}|}$, $W_a\in\mathbb{R}^{d\times|\mathsf{r}|}$, $W_o\in\mathbb{R}^{|w|\times d}$, three index sets s, r, and w, and a timestep map $K:\mathbb{Z}^+\mapsto(\mathbb{Z}^{+*})$. Given an input matrix $\boldsymbol{h}$, it outputs a matrix with entries:

$$\boldsymbol{h}_{i,\mathsf{w}}^{(l+1)}=\left(W_o\left(\left(\frac{W_a}{|K(i)|}\sum_{k\in K(i)}\boldsymbol{h}_k^{(l)}[\mathsf{r}]\right)\circledast W\boldsymbol{h}_i^{(l)}[\mathsf{s}]\right)\right)\quad i=1,...,T;\tag{26}$$

$$\boldsymbol{h}_{i,j\notin\mathsf{w}}^{(l+1)}=\boldsymbol{h}_{i,j\notin\mathsf{w}}^{(l)}\quad i=1,...,T;\tag{27}$$

We additionally require that $j\in K(i)\implies j<i$ (since self-attention is causal.)

(For simplicity, we did not include a possible bias term in linear projections $W_o, W_a, W$, we can always assume the accompanying bias parameters $\boldsymbol{b}_0, \boldsymbol{b}_a, \boldsymbol{b}$ when needed)

## C.3  REDUCING LEMMA 1 OPERATORS TO RAW OPERATOR

Given this operator, we can define each primitive in Lemma 1 using a single RAW operator, except the **mul** and **div**. Instead of the matrix multiplication operator **mul**, we will first show the dot product **dot** (a special case of **mul**), then later in the proof, we will argue that we can parallelize these dot products in Appendix C.5 to obtain **mul**. We will show how to implement div separately in Appendix C.6.

**Lemma 2.** *We can define* `mov`*,* `aff` *operator, and the dot product case of* `mul` *in Lemma 1 by using a single RAW operator*

$$\texttt{dot}(\boldsymbol{h}; (i,j), (i',j'), (i'',j'')) = \texttt{mul}(\boldsymbol{h}; 1, |i-j|, 1, (i,j), (i',j'), (i'',i''+1))$$
$$= \texttt{RAW}(\boldsymbol{h}; \odot, W=I, W_a=I, W_o=\mathbb{1}^\top, \mathsf{s}=(i,j), \mathsf{r}=(i',j'), \mathsf{w}=(i'',i''+1), K=\{(t,\{t\})\forall_t\})$$

$$\texttt{aff}(\boldsymbol{h}; (i,j), (i',j'), (i'',j''), W_1, W_2, b)$$
$$= \texttt{RAW}(\boldsymbol{h}; +, W=W_1, W_a=W_2, W_o=I, \boldsymbol{b}_0=b, \mathsf{s}=(i,j), \mathsf{r}=(i',j'), \mathsf{w}=(i'',i''+1), K=\{(t,\{t\})\forall_t\})$$

$$\texttt{mov}(\boldsymbol{h}; s,t,(i,j),(i',j'))$$
$$= \texttt{RAW}(\boldsymbol{h}; +, W=0, W_a=I, W_o=I, \mathsf{s}=(), \mathsf{r}=(i',j'), \mathsf{w}=(i,j), K=\{(t,\{s\})\})$$

*Proof.* Follows immediately by substituting parameters into Eq. (26). □

## C.4 IMPLEMENTING RAW

It remains only to show:

**Lemma 3.** *A single transformer layer can implement the* RAW *operator: there exist settings of transformer parameters such that, given an arbitrary hidden matrix* $\boldsymbol{h}$ *as input, the transformer computes* $\boldsymbol{h}'$ *(Eq. (26)) as output.*

Our proof proceeds in stages. We begin by providing specifying initial embedding and positional embedding layers, constructing inputs to the main transformer layer with necessary positional information and scratch space. Next, we prove three useful procedures for bypassing (or exploiting) non-linearities in the feed-forward component of the transformer. Finally, we provide values for remaining parameters, showing that we can implement the *Elementwise* and *Reduction* steps described above.

### C.4.1 EMBEDDING LAYERS

**Embedding Layer for Initialization:** Rather than inserting the input matrix $\boldsymbol{h}$ directly into the transformer layer, we assume (as is standard) the existence of a linear **embedding** layer. We can set this layer to pad the input, providing extra scratch space that will be used by later steps of our implementation.

We define the embedding matrix $W_e$ as:

$$W_e = \begin{pmatrix} I^{(d+1)\times(d+1)} & \boldsymbol{0} \\ \boldsymbol{0} & \boldsymbol{0} \end{pmatrix} \tag{28}$$

Then, the embedded inputs will be

$$\tilde{\boldsymbol{x}}_i = W_e \boldsymbol{x}_i = [0, \boldsymbol{x}_i, \boldsymbol{0}_{H-d-1}]^\top \tag{29}$$
$$\tilde{\boldsymbol{y}}_i = W_e \boldsymbol{y}_i = [\boldsymbol{y}_i, \boldsymbol{0}_{H-1}]^\top \tag{30}$$

**Position Embeddings for Attention Manipulation:** Implementing RAW ultimately requires controlling which position attends to which position in each layer. For example, we may wish to have layers in which each position attends to the previous position only, or in which even positions attends to other even positions. We can utilize position embeddings, $\boldsymbol{p}_i$, to control attention weights. In a standard transformer, the position embedding matrix is a constant matrix that is added to the inputs of the transformer after embedding layer (before the first layer), so the actual input to to the transformer is:

$$\boldsymbol{h}_i^0 = \tilde{\boldsymbol{x}}_i + \boldsymbol{p}_i \tag{31}$$

We will use these position embeddings to encode the timestep map K. To do this, we will use $2p$ units per layer ($p$ will be defined momentarily). $p$ units will be used to encode attention *keys* $\mathbf{k}_i$, and the other $p$ will be used to encode *queries* $\mathbf{q}_i$.

We define the position embedding matrix as follows:

$$p_i = [\boldsymbol{0}_{d+1}, \boldsymbol{k}_i^0, \boldsymbol{q}_i^0, \dots, \boldsymbol{k}_i^{(L)}, \boldsymbol{q}_i^{(L)}, \boldsymbol{0}_{H-2pT-1}]^\top \tag{32}$$

With $K$ encoded in positional embeddings, the transformer matrices $W_Q$ and $W_K$ are easy to define: they just need to retrieve the corresponding embedding values:

$$W_K^l = \begin{pmatrix} \mathbf{0} & \cdots & & & \\ & \ddots & & & \\ & I^{p \times p} & \mathbf{0}^{p \times p} & & \\ & & & \cdots & \\ & \vdots & & & \end{pmatrix} \qquad W_Q^l = \begin{pmatrix} \mathbf{0} & \cdots & & & \\ & \ddots & & & \\ & \mathbf{0}^{p \times p} & I^{p \times p} & & \\ & & & \cdots & \\ & \vdots & & & \end{pmatrix} \tag{33}$$

The constructions used in this paper rely on two specific timestep maps $K$, each of which can be implemented compactly in terms of $\boldsymbol{k}$ and $\boldsymbol{q}$:

**Case 1: Attend to previous token.** This can be constructed by setting:

$$\mathbf{k}_i = \boldsymbol{e}_i$$
$$\mathbf{q}_i = N \boldsymbol{e}_{i-1}$$

where $N$ is a sufficiently large number. In this case, the output of the attention mechanism will be:

$$\begin{aligned}
\boldsymbol{\alpha} &= \text{softmax}\left( (W_j^Q \boldsymbol{h}_i)^\top (W_j^K \boldsymbol{h}_{:i}) \right) \\
&= \text{softmax}\left( \mathbf{q}_i^\top [\boldsymbol{k}_1, \ldots, \boldsymbol{k}_i] \right) \\
&= \text{softmax}\left( [0, \ldots, N, 0] \right) \\
&= [0, \ldots, \underbrace{1}_{(i-1)}, 0]
\end{aligned}$$

**Case 2: Attend to a single token.** For simpler patterns, such as attention to a specific token:

$$K(i) = \begin{cases} \{t\} & i \geq t \\ \{\} & i < t \end{cases} \tag{34}$$

only 1 hidden unit is required. We set:

$$\boldsymbol{k}_i = \begin{cases} -N & i \neq t \\ N & i = t \end{cases}$$
$$\mathbf{q}_i = N$$

from which it can be verified (using the same procedure as in Case 1) that the desired attention pattern is produced.

**Intricacy: How can K be empty?** We can also cause $K(i)$ to attend to an empty set by assuming the softmax has extra ("imaginary") timestep obtained by prepending a 0 to attention vector pot-hoc (Chen et al., 2021).

Cumulatively, the parameter matrices defined in this subsection implement the *Read with Attention* component of the RAW operator.

### C.4.2 Handling & Utilizing Nonlinearities

The mul operator requires elementwise multiplication of quantities stored in hidden states. While transformers are often thought of as only straightforwardly implementing *affine* transformations on hidden vectors, their nonlinearities in fact allow elementwise multiplication to a high degree of approximation. We begin by observing the following property of the GeLU activation function in the MLP layers of the Transformer network:

**Lemma 4.** *The GeLU nonlinearity can be used to perform multiplication: specifically,*

$$\sqrt{\frac{\pi}{2}} \left( \text{GeLU}(x+y) - \text{GeLU}(x) - \text{GeLU}(y) \right) = xy + \mathcal{O}(x^3 + y^3) \tag{35}$$

*Proof.* A standard implementation of the GeLU nonlinearity is defined as follows:

$$\text{GeLU}(x) = \frac{x}{2}\left(1 + \tanh\left(\sqrt{\frac{2}{\pi}}\left(x + 0.044715x^3\right)\right)\right). \tag{36}$$

Thus

$$\text{GeLU}(x) = \frac{x}{2} + \sqrt{\frac{2}{\pi}}x^2 + \mathcal{O}(x^3) \tag{37}$$

$$\text{GeLU}(x + y) - \text{GeLU}(x) - \text{GeLU}(y) = \sqrt{\frac{2}{\pi}}xy + \mathcal{O}(x^3 + y^3) \tag{38}$$

$$\implies xy \approx \sqrt{\frac{\pi}{2}}(GeLU(x + y) - GeLU(x) - GeLU(y)) \tag{39}$$

For small $x$ and $y$, the third-order term vanishes. By scaling inputs down by a constant before the GeLU layer, and scaling them up afterwards, models may use the GeLU operator to perform elementwise multiplication. □

We can generalize this proof to other smooth functions as we discussed further in [TODO REF]. Previous work also shows, in practice, Transformers with ReLU activation utilize non-linearities to get the multiplication in other settings.

When implementing the `aff` operator, we have the opposite problem: we would like the output of addition to be transmitted *without* nonlinearities to the output of the transformer layer. Fortunately, for large inputs, the GeLU nonlinearity is very close to linear; to bypass it it suffices to add to inputs a large $N$:

**Lemma 5.** *The GeLU nonlinearity can be bypassed: specifically,*

$$\text{GeLU}(N + x) - N \approx x \quad N \gg 1 \tag{40}$$

*Proof.*

$$\text{GeLU}(N + x) - N = \frac{N}{2}\left(1 + \tanh\left(\sqrt{\frac{2}{\pi}}\left(N + 0.044715N^3\right)\right)\right) - N \tag{41}$$

$$\approx \frac{N}{2}(1 + 1) - N \tag{42}$$

$$= x \tag{43}$$

□

For all verions of the `RAW` operator, it is additionally necessary to bypass the LayerNorm operation. The following formula will be helpful for this:

**Lemma 6.** *Let $N$ be a large number and $\lambda$ the LayerNorm function. Then the following approximation holds:*

$$\sqrt{\frac{2}{L}}N\lambda([\boldsymbol{x}, N, -N - \sum\boldsymbol{x}, \mathbf{0}]) \approx [\boldsymbol{x}, 2N, -2N - \sum\boldsymbol{x}, \mathbf{0}] \quad N \gg 1 \tag{44}$$

*Proof.*

$$\mathbb{E}[\boldsymbol{x}] = 0 \tag{45}$$

$$\text{Var}[\boldsymbol{x}] = \frac{1}{L}(N^2 + N^2 + x^2) \approx \frac{2N^2}{L} \tag{46}$$

$$\tag{47}$$

Then,

$$\sqrt{\frac{2}{L}}N\lambda([\boldsymbol{x}, N, -N - \sum x, \mathbf{0}]) \approx \sqrt{\frac{2}{L}}N[\sqrt{\frac{L}{2N^2}}\boldsymbol{x}, \sqrt{2L}, -\sqrt{2L} - \sqrt{\frac{L}{2N^2}}\sum\boldsymbol{x}, \mathbf{0}]$$

$$= [\boldsymbol{x}, 2N, -2N - \sum\boldsymbol{x}, \mathbf{0}]$$

□

By adding a large number $N$ to two padding locations and sum the part of the hidden state that we are interested to pass through LayerNorm, we make $x$ to the output of LayerNorm pass through. This addition can be done in the transformer's feed-forward computation (with parameter $W^F$) prior to layer norm. This multiplication of $\sqrt{\frac{2}{L}}N$ can be done in first layer of MLP back, then linear layer can output/use $x$. For convenience, we will henceforth omit the LayerNorm operation when it is not needed.

We may make each of these operations as precise as desired (or allowed by system precision). With them defined, we are ready to specify the final components of the RAW operator.

### C.4.3 PARAMETERIZING RAW

We want to show a layer of Transformer defined in above, hence parameterized by $\theta = \{W_\mathrm{f}, W_1, W_2, (W^Q, W^K, W^v)_m\}$, can well-approximate the RAW operator defined in Eq. (25). We will provide step by step constructions and define the parameters in $\theta$. Begin by recalling the transformer layer definition:

$$\boldsymbol{\alpha} = \mathrm{softmax}\Big((W_j^Q \boldsymbol{h}_i)^\top (W_j^K H_{:i})\Big) \tag{48}$$

$$\boldsymbol{b}_j = \boldsymbol{\alpha}(W_j^V H_{:i}) \tag{49}$$

$$\boldsymbol{a}_i = W^F[\boldsymbol{b}_1, \ldots, \boldsymbol{b}_m] \tag{50}$$

$$\boldsymbol{h}^{(l+1)} = \mathrm{FF}(\boldsymbol{a}; W_1, W_2) \tag{51}$$

$$= W_1\sigma(W_2\lambda(\boldsymbol{a} + \boldsymbol{h}^{(l)})) + \boldsymbol{a} + \boldsymbol{h}^{(l)} . \tag{52}$$

**Attention Output** We will only use $m = 2$ attention heads for this construction. We show in Eq. (32) that we can control attentions to uniformly attend with a pattern by setting *key* and *query* matrices. Assume that the first head parameters $W_1^Q, W_1^K$ have been set in the described way to obtain the pattern function $\mathbf{K}$.

Now we will set remaining attention parameters $W_1^V, W_2^Q, W_2^K, W_2^V$ and show hat we can make the $\boldsymbol{a}_i + \boldsymbol{h}_i^{(l)}$ term in Eq. (4) to contain the corresponding term in Eq. (25), in some unused indices t such that:

$$(\boldsymbol{a}_i^{(l)} + \boldsymbol{h}_i^{(l)})_\mathsf{t} = \left(\frac{W_a}{|K(i)|} \sum_{k \in K(i)} h_k^l[\mathsf{r}]\right) \tag{53}$$

$$(\boldsymbol{a}_i^{(l)} + \boldsymbol{h}_i^{(l)})_{t' \notin \mathsf{t}} = (\boldsymbol{h}_i^{(l)})_{t' \notin \mathsf{t}} \tag{54}$$

Then the term on the RAW operator can be obtained by the first head's output. In order to achieve that, we will set $W_a$ as a part of actual attention value network such that $W_1^V$ is sparse matrix 0 everywhere expect:

$$(W_1^V)_{\mathsf{t}[m], \mathsf{r}[n]} = (W_a)_{m,n} \quad m \in 1, ..., |\mathsf{t}|, n \in 1, ..., |\mathsf{r}| \tag{55}$$

Now our first heads stores the right term in Eq. (53) in the indicies t. However, when we add the residual term $\boldsymbol{h}_i^{(l)}$, this will change. To remove the residual term, we will use another head to output $\boldsymbol{h}_i^{(l)}$, by setting $W_2^Q, W_2^K$ such that $K(i) = i$, and $W_2^V$ (similar to Eq. (34)):

$$(W_2^V)_{\mathsf{t}[m], \mathsf{r}[n]} = -1 \quad m \in 1, ..., |\mathsf{t}|, n \in 1, ..., |\mathsf{r}| \tag{56}$$

Then, $W^\mathrm{f} \in \mathbb{R}^{H \times 2H}$ is zero otherwise:

$$(W^\mathrm{f})_{\mathsf{t}[m], \mathsf{t}[m]} = 1 \quad m \in 1, ..., |\mathsf{t}| \tag{57}$$

$$(W^\mathrm{f})_{\mathsf{t}[m], \mathsf{t}[m]+H} = -1 \quad m \in 1, ..., |\mathsf{t}| \tag{58}$$

$$\tag{59}$$

We already defined $(W^Q, W^K, W^V)_{1,2}$ and $W^\mathrm{f}$ and obtained the first term in the Eq. (25) in $(\boldsymbol{a}_i + \boldsymbol{h}_i^{(l)})_{t' \in \mathsf{t}}$.

**Arithmetic term** Now we want to calculate the term inside the parenthesis Eq. (25). We will calculate it through the MLP layer and store in $\boldsymbol{m}_i$ and substract the first term. Let's denote the input to the MLP as $\boldsymbol{x}_i = (\boldsymbol{a}_i + \boldsymbol{h}_i^{(l)})$, the output of the first layer $\boldsymbol{u}_i$, the output of the non-linearity as $\boldsymbol{a}_i$, and the final output as $\boldsymbol{m}_i$. The entries of $\boldsymbol{m}_i$ will be:

$$(\boldsymbol{m}_i)_{t' \in \mathsf{w}} = W_o \left( \left( \frac{W_a}{|K(i)|} \sum_{k \in K(i)} \boldsymbol{h}_k^{(l)}[\mathsf{r}] \right) \circledast W \boldsymbol{h}_i^{(l)}[\mathsf{s}] \right) - \boldsymbol{x}_i[\mathsf{w}] \tag{60}$$

$$(\boldsymbol{m}_i)_{t' \in \mathsf{t}} = -\boldsymbol{x}_i[\mathsf{t}] \tag{61}$$

$$(\boldsymbol{m}_i)_{t' \notin (\mathsf{t} \cup \mathsf{w})} = 0 \tag{62}$$

We will define the MLP layer to operate the attention term calculated above with a part of the current hidden state by defining $W_1$ and $W_2$. Let's assume we bypass the LayerNorm by using Lemma 6.

Let's show this seperately for $+$ and $\odot$ operators.

**RAW**$(+, .)$ If the operator, $\circledast = +$, first layer of the MLP will calculate the second term in Eq. (25) and overwrite the space where the attention output term Eq. (53) is written, and add a large positive bias term $N$ to by pass GeLU as explained in Lemma 4. We will use an available space $\hat{\mathsf{t}}$ in the $x_i$ same size as $\mathsf{t}$.

$$(\boldsymbol{u}_i)_{t' \in \hat{\mathsf{t}}} = W \boldsymbol{h}_i^{l-1}[\mathsf{s}] + \boldsymbol{x}_i[\mathsf{t}] + N \tag{63}$$

$$(\boldsymbol{u}_i)_{t' \in \mathsf{t}} = -\boldsymbol{x}_i[\mathsf{t}] + N \tag{64}$$

$$(\boldsymbol{u}_i)_{t' \in \mathsf{w}} = -\boldsymbol{x}_i[\mathsf{w}] + N \tag{65}$$

$$(\boldsymbol{u}_i)_{t' \notin (\mathsf{t} \cup \hat{\mathsf{t}} \cup \mathsf{w})} = -N \tag{66}$$

This can be done by setting $W_1$ (weight term of the first layer of the MLP) to zero except the below indices:

$$(W_1)_{\hat{\mathsf{t}}[m], \mathsf{s}[n]} = (W)_{m,n} \quad m \in 1, ..., |\hat{\mathsf{t}}|, n \in 1, \ldots, |\mathsf{s}| \tag{67}$$

$$(W_1)_{\hat{\mathsf{t}}[m], \mathsf{t}[n]} = +1 \quad m \in 1, \ldots, |\hat{\mathsf{t}}|, n \in 1, \ldots, |\mathsf{t}| \tag{68}$$

$$(W_1)_{\mathsf{t}[m], \mathsf{t}[m]} = -1 \quad m \in 1, \ldots, |\mathsf{t}| \tag{69}$$

$$(W_1)_{\mathsf{w}[m], \mathsf{w}[m]} = -1 \quad m \in 1, \ldots, |\mathsf{w}| \tag{70}$$

$$\tag{71}$$

$$\tag{72}$$

and the bias vector $\boldsymbol{b}_1$ to

$$(\boldsymbol{b}_1)_{t' \in \mathsf{t}} = N \tag{73}$$

$$(\boldsymbol{b}_1)_{t' \in \mathsf{w}} = N \tag{74}$$

$$(\boldsymbol{b}_1)_{t' \in \hat{\mathsf{t}}} = N \tag{75}$$

$$(\boldsymbol{b}_1)_{t' \notin \mathsf{t} \cup \mathsf{w} \cup \hat{\mathsf{t}}} = -N \tag{76}$$

$$\tag{77}$$

Note the second term is added to make unused indices $\mathsf{t} \cup \mathsf{w} \cup \hat{\mathsf{t}}$ become zero after the gelu, which outputs zero for large negative values. Since we added a large positive term, we make sure gelu behaved like a linear layer. Thus we have,

$$(\boldsymbol{v}_i)_{t' \in \hat{\mathsf{t}}} = W \boldsymbol{h}_i^l[\mathsf{s}] + \boldsymbol{x}_i[\mathsf{t}] + N \tag{78}$$

$$(\boldsymbol{v}_i)_{t' \in \mathsf{t}} = -\boldsymbol{x}_i[\mathsf{t}] + N \tag{79}$$

$$(\boldsymbol{v}_i)_{t' \in \mathsf{w}} = -\boldsymbol{x}_i[\mathsf{w}] + N \tag{80}$$

$$(\boldsymbol{v}_i)_{t' \notin \mathsf{t} \cup \mathsf{w} \cup \hat{\mathsf{t}}} = 0 \tag{81}$$

Now, we need to set $W_2$, to simulate $W_o \in \mathbb{R}^{|w| \times |t|}$,

$$(W_2)_{\mathsf{w}[m], \hat{\mathsf{t}}[n]} = (W_o)_{m,n} \quad m \in 1, ..., |\mathsf{w}|, n \in 1, ..., |\mathsf{t}| \tag{82}$$

$$(W_2)_{\mathsf{t}[m], \mathsf{t}[m]} = +1 \quad m \in 1, \ldots, |\mathsf{t}| \tag{83}$$

$$(W_2)_{\mathsf{w}[m], \mathsf{w}[m]} = +1 \quad m \in 1, \ldots, |\mathsf{w}| \tag{84}$$

$$\tag{85}$$

$$(b_2)_{\mathsf{w}[m]} = -N \sum_j (W_o)_{m,j} - N \quad m \in 1, \ldots, |\mathsf{w}| \tag{86}$$

$$(b_2)_{\mathsf{t}[i]} = -N \tag{87}$$

$$(b_2)_{t' \notin \mathsf{t}} = 0 \tag{88}$$

$$\tag{89}$$

Therefore, $m_i[w] = W_o x_i[\mathsf{t}] + W_0 W h_i^l[\mathsf{s}] - x_i[\mathsf{w}]$ equals to what we promised in Eq. (60) for + case. If we sum this with the residual $x_i$ term back Eq. (53), so the output of this layer can be written as:

$$(h_i)_{t' \in \mathsf{w}}^{(l+1)} = W_o \left( \left( \frac{W_a}{|K(i)|} \sum_{k \in K(i)} h_k^{(l)}[\mathsf{r}] \right) + W h_i^{(l)}[\mathsf{s}] \right) \tag{90}$$

$$(h_i)_{t' \notin \mathsf{w}}^{(l+1)} = (h_i^l)_{t' \notin \mathsf{w}} \tag{91}$$

**RAW**$(\odot, .)$  If the operator, $\circledast = \odot$, we need to use three extra hidden units the same size as $|\mathsf{t}|$, let's name the extra indices as $\mathsf{t}_a$, $\mathsf{t}_b$, $\mathsf{t}_c$, and output $w$ space. The $(u_i)$ will get below entries to be able to use [], where $N$ is a large number:

$$(u_i)_{t' \in \mathsf{t}_a} = (W h_i^l[\mathsf{s}] + x_i[\mathsf{t}])/N \tag{92}$$

$$(u_i)_{t' \in \mathsf{t}_b} = x_i[\mathsf{t}]/N \tag{93}$$

$$(u_i)_{t' \in \mathsf{t}_c} = W h_i^l[\mathsf{s}]/N \tag{94}$$

$$(u_i)_{t' \in \mathsf{t}} = -x_i[\mathsf{t}] + N \tag{95}$$

$$(u_i)_{t' \in \mathsf{w}} = -x_i[\mathsf{w}] + N \tag{96}$$

$$(u_i)_{t' \notin (\mathsf{t} \cup \mathsf{t}_a \cup \mathsf{t}_b \cup \mathsf{t}_c \cup \mathsf{w})} = -N \tag{97}$$

$$\tag{98}$$

All of this operations are linear, can be done $W_1$ zero except the below entries:

$$(W_1)_{\mathsf{t}_a[m],\mathsf{s}[n]} = (W)_{m,n}/N \quad m \in 1, ..., |\mathsf{t}_a|, n \in 1, ..., |\mathsf{s}| \tag{99}$$

$$(W_1)_{\mathsf{t}_a[m],\mathsf{t}[n]} = 1/N \quad m \in 1, ..., |\mathsf{t}_a|, n \in 1, ..., |\mathsf{t}| \tag{100}$$

$$(W_1)_{\mathsf{t}_b[m],\mathsf{t}[m]} = 1/N \quad m \in 1, ..., |\mathsf{t}_b|, n \in 1, ..., |\mathsf{t}| \tag{101}$$

$$(W_1)_{\mathsf{t}_c[m],\mathsf{s}[m]} = 1/N \quad m \in 1, ..., |\mathsf{t}_c|, n \in 1, ..., |\mathsf{s}| \tag{102}$$

$$(W_1)_{\mathsf{w}[m],\mathsf{w}[m]} = -1 \quad m \in 1, ..., |\mathsf{w}| \tag{103}$$

$$(W_1)_{\mathsf{t}[i],\mathsf{t}[m]} = -1 \quad m \in 1, ..., |\mathsf{t}| \tag{104}$$

$$\tag{105}$$

and $b_1$ to:

$$(b_1)_{t' \in (\mathsf{t} \cup \mathsf{t}_a \cup \mathsf{t}_b \cup \mathsf{t}_c)} = 0 \tag{106}$$

$$(b_1)_{t' \in (\mathsf{t} \cup \mathsf{w})} = N \tag{107}$$

$$(b_1)_{t' \notin (\mathsf{t} \cup \mathsf{t}_a \cup \mathsf{t}_b \cup \mathsf{t}_c \cup \mathsf{w})} = -N \tag{108}$$

$$\tag{109}$$

The resulting $v$ with the approximations become:

$$(v_i)_{t' \in \mathsf{t}_a} = \mathrm{gelu}((W h_i^l[\mathsf{s}] + x_i[\mathsf{t}])/N) \tag{110}$$

$$(v_i)_{t' \in \mathsf{t}_b} = \mathrm{gelu}(x_i[t]/N) \tag{111}$$

$$(v_i)_{t' \in \mathsf{t}_c} = \mathrm{gelu}(W h_i^l[s]/N) \tag{112}$$

$$(v_i)_{t' \in \mathsf{t}} = x_i[\mathsf{t}] + N \tag{113}$$

$$(v_i)_{t' \in \mathsf{w}} = x_i[\mathsf{w}] + N \tag{114}$$

$$(v_i)_{t' \notin (\mathsf{t} \cup \mathsf{t}_a \cup \mathsf{t}_b \cup \mathsf{t}_c \cup \mathsf{w})} = 0 \tag{115}$$

$$\tag{116}$$

Now, we can use the GeLU trick in Lemma 4, by setting $W_2$

$$(W_2)_{\mathsf{w}[m],\mathsf{t}_a[n]} = (W_o)_{m,n}N^2\sqrt{\frac{\pi}{2}} \quad m \in 1,\ldots,|\mathsf{w}|, n \in 1,\ldots,|\mathsf{t}_a| \tag{117}$$

$$(W_2)_{\mathsf{w}[m],\mathsf{t}_b[n]} = -(W_o)_{m,n}N^2\sqrt{\frac{\pi}{2}} \quad m \in 1,\ldots,|\mathsf{w}|, n \in 1,\ldots,|\mathsf{t}_b| \tag{118}$$

$$(W_2)_{\mathsf{w}[m],\mathsf{t}_c[n]} = -(W_o)_{m,n}N^2\sqrt{\frac{\pi}{2}} \quad m \in 1,\ldots,|\mathsf{w}|, n \in 1,\ldots,|\mathsf{t}_c| \tag{119}$$

$$(W_2)_{\mathsf{w}[m],\mathsf{w}[m]} = 1 \quad m \in 1,\ldots,|\mathsf{w}| \tag{120}$$

$$(W_2)_{\mathsf{t}[m],\mathsf{t}[m]} = 1 \quad m \in 1,\ldots,|\mathsf{t}| \tag{121}$$

$$\tag{122}$$

We then set $b_2$:

$$(\boldsymbol{b}_2)_{t' \in (t\cup\mathsf{w})} = N \tag{123}$$

$$(\boldsymbol{b}_2)_{t' \notin (t\cup\mathsf{w})} = 0 \tag{124}$$

$$\tag{125}$$

With this, $\boldsymbol{m}_i[w] = W_o\boldsymbol{x}_i[t] * W_0 W \boldsymbol{h}_i^{l-1}[s] - \boldsymbol{x}_i[w]$, and

$$(\boldsymbol{h}_i)_{t' \in \mathsf{w}}^{(l+1)} = W_o\left(\left(\frac{W_a}{|K(i)|}\sum_{k \in K(i)} \boldsymbol{h}_k^{(l)}[\mathsf{r}]\right) \odot W\boldsymbol{h}_i^{(l)}[\mathsf{s}]\right) \tag{126}$$

$$(\boldsymbol{h}_i)_{t' \notin \mathsf{w}}^{(l+1)} = (\boldsymbol{h}_i^l)_{t' \notin \mathsf{w}} \tag{127}$$

We have used $4|t|$ space for internal computation of this operation, and finally used $|\mathsf{w}|$ space to write the final result. We show RAW operator is implementable by setting the parameters of a Transformer.

## C.5 PARALLELIZING THE RAW OPERATOR

**Lemma 7.** *With the conditions that $K$ is constant, the operators are independent (i.e $(r_i \cup s_i \cup w_i) \cap w_{j \neq i} = \emptyset$), and there is $\sum_k(4|\mathsf{t}_k| + |\mathsf{w}_k|)$ available space in the hidden state, then a Transformer layer can apply $k$ such* RAW *operation in parallel by setting different regions of $W_1, W_2, W_f$ and $(W^V)_k$ matrices.*

*Proof.* From the construction above, it is straightforward to modify the definition of the RAW operator to perform $k$ operations as all the indices of matrices that we use in Appendix C.4.3 do not overlap with the given conditions in the lemma. □

This makes it possible to construct a Transformer layer not only to implement vector-vector dot products, but general matrix-matrix products, as required by **mul**. With this, we show that we can implement **mul** by using single layer of a Transformer.

## C.6 LAYERNORM FOR DIVISION

Let say we have the input $[c, \boldsymbol{y}, \boldsymbol{0}]^\top$ calculated before the attention output in Eq. (53), and we want to divide $\boldsymbol{y}$ to $c$. This trick is very similar to the on in Lemma 6. We can use the following formula:

**Lemma 8.** *using LayerNorm for division. Let $N, M$ to be large numbers, $\lambda$ LayerNorm function, the following approximation holds:*

$$\sqrt{\frac{2}{L}}MN\lambda([Nc, \frac{\boldsymbol{y}}{M}, -Nc - \sum\frac{\boldsymbol{y}}{M}, \boldsymbol{0}]) \approx [MN, \frac{\boldsymbol{y}}{c}, -MN - \frac{\boldsymbol{y}}{c}, \boldsymbol{0}] \tag{128}$$

*Proof.*

$$\mathbb{E}[\boldsymbol{x}] = 0 \tag{129}$$

$$\mathrm{Var}[\boldsymbol{x}] = \frac{1}{L}\left(N^2c^2 + \frac{1}{M}\sum\boldsymbol{y}^2 + \left(Nc + \sum\frac{\boldsymbol{y}}{M}\right)^2\right) \approx \frac{2N^2c^2}{L} \tag{130}$$

$$\tag{131}$$

Then,

$$\sqrt{\frac{2}{L}} MN\lambda([Nc, \frac{\boldsymbol{y}}{M}, -Nc - \sum \frac{\boldsymbol{y}}{M}, \boldsymbol{0}]) = \sqrt{\frac{2}{L}} MN[\sqrt{\frac{L}{2}}, \sqrt{\frac{L}{2}}\frac{\boldsymbol{y}}{MNc}, -\sqrt{\frac{L}{2}} - \sqrt{\frac{L}{2}}\sum \frac{\boldsymbol{y}}{MNc}, 0]$$

$$= [MN, \underbrace{\frac{\boldsymbol{y}}{c}}_{\text{wanted result}}, -MN - \frac{\boldsymbol{y}}{c}, \boldsymbol{0}]$$

$\square$

To get the input to the format used in this Lemma, we can easily use $W_f$ to convert the head outputs. Then, after the layer norm, we can use $W_1$ to pull the $\frac{\boldsymbol{y}}{c}$ back and write it to the attention output. By this way, we can approximate scalar division in one layer.

**Lemma 1** By Lemmas 2, 3, 3, 7 and 8; we constructed the operators in Lemma 1 using single layer of a Transformer, thus proved Lemma 1 $\square$

## D  DETAILS OF TRANSFORMER ARHITECTURE AND TRAINING

We perform these experiments using the Jax framework on P100 GPUs. The major hyperparameters used in these experiments are presented in Table 1. The code repository used for reproducing these experiments will be open sourced at the time of publication. Most of the hyperparameters adapted from previous work Garg et al. (2022) to be compatible, and we adapted the Transformer architecture details. We use Adam optimizer with cosine learning rate scheduler with warmup where number of warmup steps set to be 1/5 of total iterations. We use larned absolute position embeddings.

| Parameter | Search Range |
|---|---:|
| Number of heads | 1, 2, **4**, 8 s |
| Number of layers | 1, 2, 12, **16** |
| Hidden size | 16, 32, 64, 256, **512**, 1024 |
| Batch size | 64 |
| Maximum number of epochs | 500.000 |
| Initial Learning rate ($lr_i$) | **1e-4**, 2.5e-4 |
| Weight decay | **0**, 1e-5 |
| Bias initialization | **uniform scaling**, normal(1.0) |
| Weight initialization | **uniform scaling**, normal(1.0) |
| Position embedding initialization | **uniform scaling**, normal(1.0) |

Table 1: Hyperparameters used in the ICL. The best parameter for each hyperparameter is highlighted.

In the phase shift plots in Fig. 3, we keep the value in the x-axis constant and used the best setting over the parameters: {number of layers, hidden size, number of heads and learning rate}.

## E  DETAILS OF PROBE

We will use the terms *probe model* and *task model* to distinguish probe from ICL. Our probe is defined as:

$$\boldsymbol{\alpha} = \text{softmax}(\boldsymbol{s}_v) \tag{132}$$

$$\hat{\boldsymbol{v}} = \text{FF}_v(\boldsymbol{\alpha}^\top W_v \boldsymbol{h}) \tag{133}$$

The position scores $\boldsymbol{s}_v \in \mathbb{R}^T$ are learned parameters where $T$ is the max input sequence length ($T = 80$ in our experiments). The softmax of position scores attention weights $\boldsymbol{\alpha}$ for each position and for each target variable. This enables us to learn input-independent, optimal target locations for each target (displayed on the right side of Fig. 4). We then average hidden states by using these attention weights. A linear projection, $W_v \in \mathbb{R}^{T \times H'}$, is applied before averaging. FF is either a linear layer or a 2-layer MLP (hidden size=512) with a GeLU activation function. For each layer, we train a different probe with different parameters using stochastic gradient descent. $H'$ equals to the 512. The probe is trained using an Adam optimizer with a learning rate of 0.001 (chosen from among $\{0.01, 0.001, 0.0001\}$ on validation data).

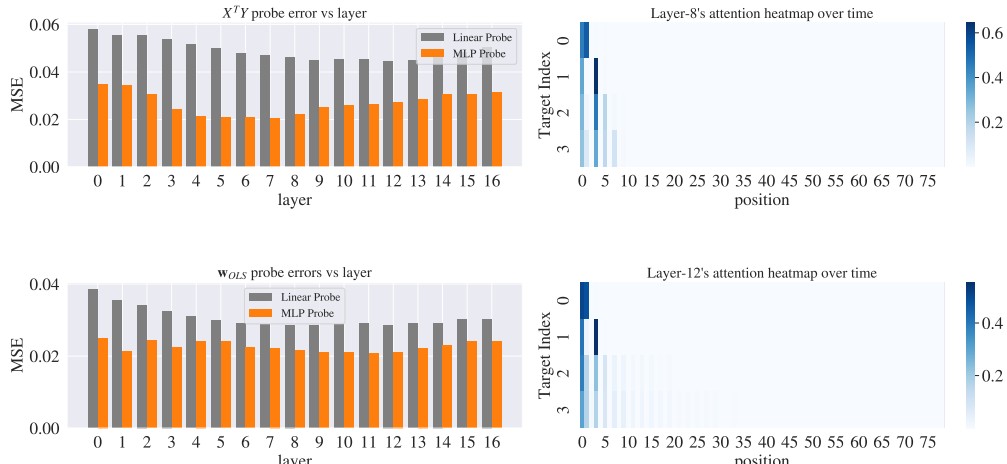

Figure 5: Detailed error values of the control probe displayed in Fig. 4.

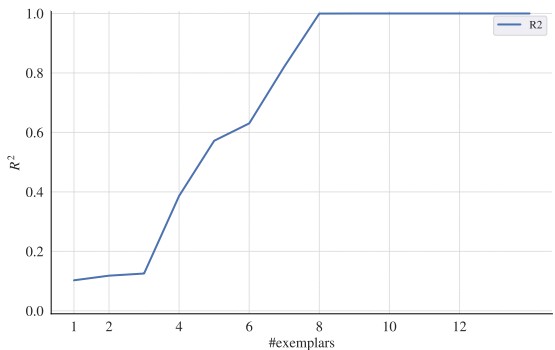

Figure 6: $R^2$ of linear weight estimation on $d = 8$ problem

**Control Experiments**   In Fig. 4, dashed lines show probing results with a task model trained on a *control task*, in which $w$ is always the all-ones $\mathbf{1}$. This problem structurally resembles our main experiment setup, but does not require in-context learning. During probing, we feed this model data generated by $w$ sampled form normal distribution as in the original task model. We observe that the control probe has a significantly higher error rate, showing that the probing accuracy obtained with actual task model is non-trivial. We present detailed error values of the control probe in Fig. 5.

## F  LINEARITY OF ICL

In Fig. 1b, we compare implicit linear weight of the ICL against the linear algorithms using ILWD measure. Note that this measure do not assume predictors to be linear: when the predictors are not linear, ILWD measures the difference between closest linear predictors (in Eq. (16) sense) to each algorithm.

To gain more insight to ICL's algorithm, we can measure how linear ICL in different regimes of the linear problem (underdetermined, determined) by using $R^2$ (coefficient of determination) measure. So, instead of asking what's the best linear fit in Eq. (16), we can ask how good is the linear fit, which is the $R^2$ of the estimator. Interestingly, even though our model matches min-norm least square solution in both metrics in Section 4.3, we show that ICL is becoming gradually linear in the under-determined regime Fig. 6. This is an important result, enables us to say the in-context learner's hypothesis class is not purely linear.

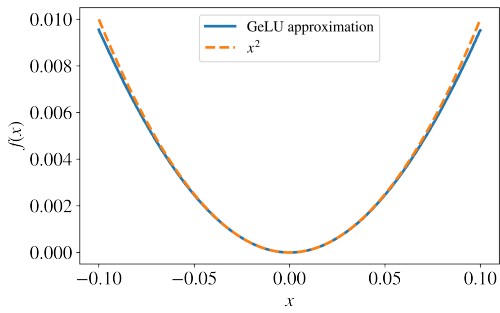

(a) Approximating $x^2$ using GeLU , Eq. (136).

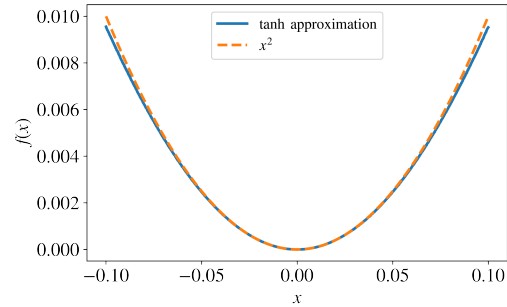

(b) Approximating $x^2$ using $\tanh$, Eq. (140), where $\delta = 1e^{-3}$.

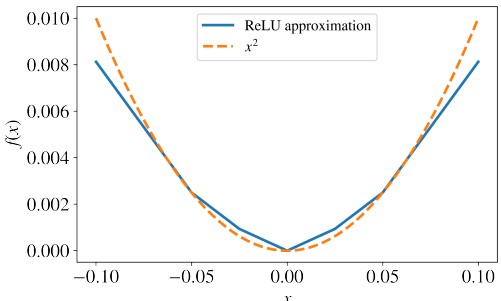

(c) A piece-wise linear approximation to $x^2$ by using ReLU, Eq. (141).

Figure 7: Approximations of multiplication via various non-linearities.

## G  MULTIPLICATIVE INTERACTIONS WITH OTHER NON-LINEARITIES

We can show that for a real-valued and smooth non-linearity $f(x)$, we can apply the same trick in in the paper body. In particular, we can write Taylor expansion as:

$$f(x) = \sum_{i=0}^{\infty} a_i x^i = a_0 + a_1 x + a_2 x^2 + \dots \tag{134}$$

which converges for some sufficiently small neighborhood: $\mathcal{X} \in [-\epsilon, \epsilon]$. First, assume that the second order term $a_2$ dominates higher-order terms in this domain such that:

$$a_2 x^2 \gg a_{i>2} x^i \quad \text{where} x \in \mathcal{X}$$

.

It's is easy to verify that the following is true:

$$\frac{1}{2a_2} \left( f(x+y) - f(x) - f(y) + a_0 \right) = xy + \mathcal{O}(x^3 + y^3) \tag{135}$$

So, given the expansion for GeLU in Eq. (37), we can use this generic formula to obtain the multiplication approximation:

$$\sqrt{\frac{\pi}{2}} (GeLU(x+y) - GeLU(x) - GeLU(y)) \approx xy \tag{136}$$

We plot this approximation against $x^2$ for $[0.1, -0.1]$ range in Fig. 7a.

In the case of $a_2$ is zero, we cannot get any second order term, and in the case of $a_2$ is negligible $\mathcal{O}(x^3 + y^3)$ will dominate the Eq. (135), so we cannot obtain a good approximation of $xy$. In this case, we can resort to numerical derivatives and utilize the $a_3$ term:

$$f'(x) = a_1 + 2a_2 x + 3a_3 x^3 + \dots \tag{137}$$

If $a_3$ is not negligible, $a_3 x^2 \ll a_{i>3} x^i$ in the same domain, we can use numerical derivatives to get a multiplication term:

$$\frac{1}{6a_3} \left( \frac{f(x+y+\delta) - f(x+y)}{\delta} - \frac{f(x+\delta) - f(x)}{\delta} - \frac{f(y+\delta) - f(y)}{\delta} + a_1 \right) = xy + \mathcal{O}(x^3 + y^3) \tag{138}$$

For example, $\tanh$ has no second order term in its Taylor expansion:

$$\tanh x = x - \frac{x^3}{3} + \frac{2x^5}{15} + \dots \tag{139}$$

Using above formula we can obtain the following expression:

$$-\frac{1}{2} \left( \frac{\tanh(x+y+\delta) - \tanh(x+y)}{\delta} - \frac{\tanh(x+\delta) - \tanh(x)}{\delta} \right.$$
$$\left. - \frac{\tanh(y+\delta) - \tanh(y)}{\delta} + 1 \right) \approx xy \tag{140}$$

Similar to our construction in Eq. (110), we can construct a Transformer layer that calculates these quantities (noting that $\delta$ is a small, input-independent scalar).

We plot this approximation against $x^2$ for $[0.1, -0.1]$ range in Fig. 7b. Note that, if we use this approximation in our constructions we will need more hidden space as there are 6 different $\tanh$ term as opposed to 3 GeLU term in Eq. (110).

**Non-smooth non-linearities**  ReLU is another commonly used non-linearity that is not differentiable. With ReLU, we can only hope to get piece-wise linear approximations. For example, we can try to approximate $x^2$ with the following function:

$$0.0375 * \text{ReLU}(x) + 0.0375 * \text{ReLU}(-x) + \text{ReLU}(0.05 * (x - 0.05)) +$$
$$\text{ReLU}(-0.05 * (x + 0.05)) + \text{ReLU}(0.025 * (x - 0.025)) + \text{ReLU}(-0.025 * (x + 0.025)) \approx x^2 \tag{141}$$

We plot this approximation against $x^2$ for $[0.1, -0.1]$ range in Fig. 7c.

## H  EMPIRICAL SCALING ANALYSIS WITH DIMENSIONALITY

In Figs. 3a and 3b, we showed that ICL needs different hidden sizes to enter the "Ridge regression phase" (orange background) or "OLS phase" (green background) depending on the dimensionality $d$ of inputs $x$. However, we cannot reliably read the actual relations between size requirements and the dimension of the problem from only two dimensions. To better understand size requirements, we ask the following empirical question for each dimension: how many layer/hidden size/heads are needed to better fit the least-squares solution than the $\text{Ridge}(\lambda = \epsilon)$ regression solution (the green phase in Figs. 3a and 3b)?

To answer this important question, we experimented with $d = \{1, 2, 4, 8, 12, 16, 20\}$ and run an experiment sweep for each dimension over:

- number of layers (L): $\{1, 2, 4, 8, 12, 16\}$,
- hidden size (H): $\{16, 32, 64, 256, 512, 1024\}$,
- number of heads (M): $\{1, 2, 4, 8\}$,
- learning rate: $\{1\text{e-}4, 2.5\text{e-}4\}$.

For each feature that affects computational capacity of transformer ($L$, $H$, $M$), we optimize other features and find the minimum value for the feature that satisfies $\text{SPD}(\text{OLS}, \text{ICL}) < \text{SPD}(\text{Ridge}(\lambda = \epsilon), \text{ICL})$. We plot our experiment with $\epsilon = 0.1$ in Appendix H. We find that single head is enough for all problem dimensions, while other parameters exhibit a step-function-like dependence on input size.

Please note that other hyperparameters discussed in Appendix D (e.g weight initialization) were not optimized for each dimension independently.

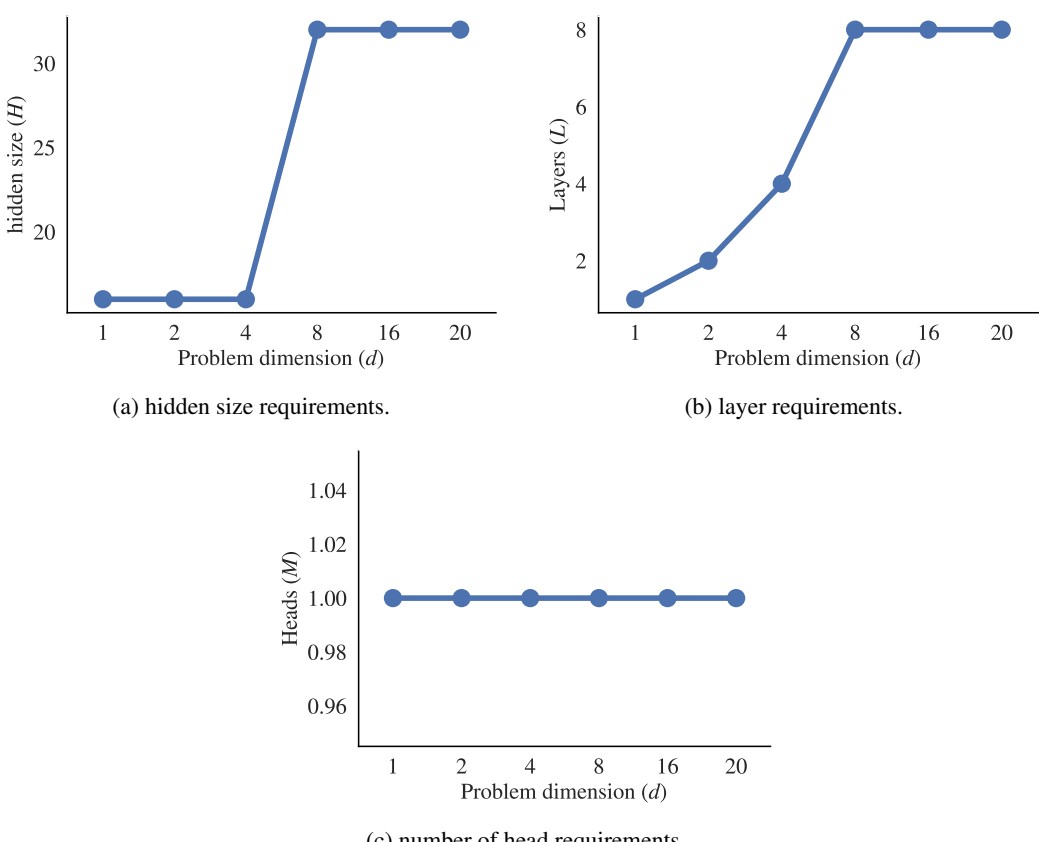

(a) hidden size requirements.

(b) layer requirements.

(c) number of head requirements.

Figure 8: Empirical requirements on model parameters to satisfy $\text{SPD}(\text{Ridge}(\lambda = 0.1), \text{ICL}) > \text{SPD}(\text{OLS}, \text{ICL})$ when other parameters optimized.

