# OpenReview forum: "​​What learning algorithm is in-context learning? Investigations with linear models"
_ICLR.cc/2023/Conference — ICLR 2023 notable top 5%_

### Official Review · Reviewer_NUsG · 2022-10-23

**Confidence:** 4
**Correctness:** 3
**Technical Novelty And Significance:** 4
**Empirical Novelty And Significance:** 4
**Recommendation:** 8

**Clarity, Quality, Novelty And Reproducibility:**

**QUALITY:** This is a high-quality submission that tackles an important problem and offers interesting and nontrivial insights.

**CLARITY:** The paper is very well-written and clear. Here are a couple additional comments (BUNCH OF REQUESTS):
* The first sentence of the second paragraph of the introduction imply that _all_ types of in-context learning can be understood via the framework discussed in the paper. It'd be better if the authors clarify that there exist other types of in-context learning (e.g. the one facilitated by induction heads), or change wording.
* The "d" that appears in the bounds (e.g. "with O(d) layers, they can compute ...") isn't quite clear. Perhaps clarifying what this "d" stands for as early on in the introduction as possible would improve clarity.
* Could you specify in the layernorm equation (Eq 7) whether the first and second moments are computed using activations from all timesteps, or computed separately for each timestep?
* The legend of Figure 1 (b) is a big confusing, as the legend on the Figure (a) doesn't, for example, have a solid black line labelled.
* **Equation for ILW:** The expectation seems to be under $D$. What about $D_A$? How does that factor in the equation? Do you have a fixed $D_A$, or do you resample it every time you sample a $D$? (I presume it's the latter, but just wanted to make sure.)
* **One step gradient descent:** Do you initialize the first guess with zeros, or do you actually initialize with noise?
* **Making figures color-blind friendly:** (not urgent until camera-ready unless there's a colorblind reviewer) would it be possible to make the plots colorblind-friendly (or at least add equivalent figures in the appendix for this purpose)?

**ORIGINALITY**: I believe the questions asked, as well as the insights provided are for the most part original/novel.


**TYPOS:**  The paper has only a few typos. Here are a couple, in case it helps the authors.
* 3rd paragraph of Introduction: "... between different predictors as model depth __and__ training set ..."
* Between equation 2 and 3: "where each \vb is is"
* Theorem 1 and 2: There's a repeated "the" in the final sentence.
* Image typo: The x axis of the right top plot in Figure 4 is half-occluded.



**Strength And Weaknesses:**

(The requests from the authors during the rebuttal phase is tagged below.)

**STRENGTHS:**
* **Scope and relevance:** Given the surge of interest in in-context learning, this paper is very timely in taking on a very important topic.
* **Significance of contributions:** The paper does a great job illustrating how a transformer might implement an iterative optimizer in its weights. While I expect this work not to be the final word in how a transformer architecture might learn to be a linear function approximator, the questions asked by the authors seem to be the right ones and the provided answers seem novel and meaningful.
* **Clarity:** The main body of the paper is written very well.
* **"RAW" operator is useful:** While not discussed in the main body of the paper, the raw operator defined and used in the appendix for the constructions seem like a quite neat way to think about constructive proofs for transformers.

**WEAKNESSES:**
* **Dependence on the GELU activations:** If I'm not mistaken, the authors make use of the numerics of the GELU activation to show how a the feed-forward layers can implement the dot product and matrix multiplication operations. I have two concerns about this:
  * It's not clear how small the activations need to be for the approximation to hold. (REQUEST) Would it be possible to add a plot in the appendix that shows how well $x * x = x^2$ is approximated with the GELU approximation? The x-axis could be the values between $-0.1$ to $+0.1$, and the y-axis would be the error in the approximation, or the proportion of the error to the x value.
  * Transformers that don't use this activation also display ICL capabilities, including the ability to be linear function approximators (happy to be corrected about this - perhaps GELU is much more important than I anticipate).
  * In light of this, I'm not sure if the construction of $mul$ truly represent what the transformers might be learning (happy to be convinced otherwise during the rebuttal) . (REQUEST) Perhaps it'd be good to at least briefly discuss the dependence on GELU to get the $mul$ construction in the discussion section following Theorem 2?
* **Mechanistic interpretability results lack control groups:** As the authors acknowledge, interpreting probing results is often tough. Hence, the results presented in Section 5 appear to be speculative as of now. (This limitation is acknowledged by the authors). (REQUEST) One approach to strengthen the results would be to produce Figure 4 (left and right) with randomly initialized, slightly trained and fully trained (but poorly performing in ICL) transformers. If these also yield comparable loss values, then discussion of the results of Section 5 should be reconsidered.
* **Making Figure 2 denser to confirm a claim:** The authors claim that for all values of $\sigma^2$ and $\tau^2$, the right parameter that gives the best fit also is the one that minimizes Bayes risk. Figure 2, which seems to be what this sentence is based on, is a bit too sparse to really reach this conclusion. If it's possible, it would be great to make the plots a bit denser to see the trends a bit more clearly. That is, sample tau and sigma a bit denser to add more measurement points? Marking the MSPD between the transformer and the predictor using the optimal ridge parameter with a different marker for each value $\tau$ and $\sigma$ on the plots could also bring extra clarity perhaps? (nitpick) Lastly, perhaps a colormap that has a gradient determined by the value of the ridge parameter could also make the plot easier to read.
* **Ambiguity in Figure 4 (right):** Figure 4 (right) isn't very clear to me. What are the x and y axes showing?


**QUESTIONS TO AUTHORS:** (a bunch of REQUESTS):
* **Position embeddings:** What kind of position embeddings did you use? (REQUEST) Space permitting, could you briefly discuss how you expect the choice of position embeddings to affect ICL result? If I'm not mistaken, the derivations in the appendix assume absolute position embeddings a'la the original transformer paper.
* **Gradient descent with n steps as another benchmark predictor:** I was wondering if you considered observing whether a few steps of gradient descent (not just a single step) matches ICL performance of medium-shallow transformers? If the initial weights are initialized with zeros, gradient descent will presumably consistently lead to an increase in the norm of the implied weight vector. Since having an L2 penalty on the weight vector also has a similar effect, perhaps this is why ICL performance of medium-shallow transformers are matched well with the predictors with a ridge term. (Analogously, perhaps gradient descent with a couple of steps has low MSPD with ridge regression with an appropriately chosen lambda?)
* **MLP used in the probe experiments:** Could you give further details regarding the MLP probe used in Section 5?


**Summary Of The Paper:**

Previous work has shown that transformers can be trained to be "in-context learners (ICL) of linear functions”. That is, given a couple of (x, y) pairs sampled from a line (that is also sampled randomly at test time), and a new x' from the same line in its context, a transformer can be trained to output the correct y'.

This work is focused on understanding *how* this happens.

This paper provides three pieces of evidence for the hypothesis that transformers in fact learn standard learning algorithms to acquire the skill described above.
1. Construction: The authors provide a bottom-up construction demonstrating how a transformer can represent both a single step of gradient descent and a single step of Sherman-Morrison update in O(1) layers. This implies these steps can be composed in  a single model to “run optimization” during the forward pass.
2. What “computation” an in-context learning transformer displays by testing how similar its outputs on held-out data is to well-known algorithms: The authors systematically show that 1) the best performing model behaves very similar to ordinary least squares when no noise is present, 2) when noise is added, the best model learns a solution that minimizes Bayes risk (very interesting). 3) Transformers that have their capacity restricted interpolate between different predictors.
3. How learned in-context-learning transformers actually implement ICL: The authors present preliminary evidence, using auxiliary probes, that the activations of ICL transformers might extract the moment matrix and the parameter vector during the forward pass.

In addition to these pieces of evidence, the authors also propose a neat way of thinking about how a transformer might represent algorithmic operations (named the "raw" operation in the appendix), which might be useful in analyses in other domains.

**Summary Of The Review:**

The authors provide a constructive, behavioural and mechanistic account of how transformer networks can in theory and practice learn to in-context-learn linear functions.

The paper is well-written with insightful questions and clear answers. I believe this is a good paper that deserves to be highlighted in NeurIPS.

---

> ### Author Response · Authors · 2022-11-15
> **Thank you!**
>
> Thank you for the positive recommendation and detailed feedback!
>
> We have worked towards implementing all of the suggestions in the review, which we believe have significantly strengthened the paper—we appreciate your help.
>
> **How good is GeLU approximation?**
> Please see the new Figure 7 in Appendix G, we visualize $x^2$ and the GeLU approximation together in the suggested range.
>
> **Is the implementation of “mul” operator specific to GeLU?**
> This is a great question! In the new Appendix G, we generalize our construction for multiplication operations to other smooth non-linearities. We provide an example that shows how to approximate multiplication with tanh function as well.
>
> **Probing experiments lacks control group**
> We thank you for this helpful suggestion. In Figure 4, we ran an additional experiment where we trained the same task to perform linear regression with a *fixed* parameter vector $w=1$ — a problem that does not require in-context learning. We show that the minimum probe error obtained with this control model is at least 2x higher than the error obtained by the original task model, indicating that probing results are not explained by the input format and model architecture alone.
>
> **Improving Figure 2**
> Thanks for the suggestion—we have implemented it for the revised version of Fig 2.
>
> **Ambiguity in Figure 4**
> We clarified the caption to explain the heatmap. We fixed Eq. 18, which actually calculates input-independent attention weights to localize targets. Please also see the revised Appendix E for more clarification about the experiment.
>
>
> **Gradient descent with n steps as another benchmark predictor**
> As you noted, because gradient descent has implicit regularization property (https://arxiv.org/pdf/1710.10345.pdf), this estimator will be similar to a Ridge regression. However, we added n-step SGD with batch size=1 to the plots in the new version, and we clarified what we mean by GD and SGD in Section 4.3.
>
> **Position embeddings**
> We use learned absolute position embeddings, and we updated Appendix D to clarify this.
>
>
> **Clarification for MLP Probe**
> It’s a 2-layer MLP with hidden size 512, and the activation is GeLU. We updated Appendix E with more details.
>
>
> **Explain the $d$ more carefully in the text**
> We agree. We clarified the paragraphs in which we use $d$ for dimensionality of the underlying problem.
>
>
> **Equation of ILW**
> This is a good catch! Indeed, an expectation of $D_{\mathcal{A}}$ (hence $D_x$) was missing. We updated the equation.
>
>
> **Initialization in gradient descent**
> We initialized it to zero as we know that our data distribution comes from zero-mean.
>
>
> **Making figures color-blind friendly**
> We’re sorry for this mistake. We had tried to use similar colors for the same class of algorithms. Therefore, we have manually picked these colors. We will fix them in the camera.
>
> **Typos and Clarifications**
> We fixed the typos mentioned in the comment.
> We change the first sentence of the second paragraph of the introduction.
> Layer norm is independent over all time steps. To clarify this, we added subscript $i$ to the Equations 4 and 5.

---

> > ### Comment · Reviewer_NUsG · 2022-11-18
> > **Thank you for your response**
> >
> > Thank you for your clarifications -- it's especially nice to see that GeLU is not strictly needed for the construction.
> >
> > I have one follow-up question. It's not needed for the rebuttal, but I think it'd be a nice to know the answer to the following question: Is it as easy to elicit in-context learning (of linear functions) using ReLU as it is with GeLU? If you've done this experiment, did you see any qualitative difference between the training curves and the end network? (Again, given how little time there's left in the rebuttal, I certainly don't expect the authors to run new experiments at this point -- just asking in case this experiment is already conducted.)

---

### Official Review · Reviewer_5hCi · 2022-10-25

**Confidence:** 4
**Correctness:** 1
**Technical Novelty And Significance:** 3
**Empirical Novelty And Significance:** Not applicable
**Recommendation:** 8

**Clarity, Quality, Novelty And Reproducibility:**

The paper is well-structured, clear and reads easily. The level of technical details provided in the main text is good. The supplementary material provides an extensive and very lengthy details on the derivation of different operators.  The paper has a placeholder link for the code, so I think that the results should be easily reproducible.


**Strength And Weaknesses:**

Strength
- The paper provides theoretical insights on how ICL implements learning algorithms. This can be used to introduce new paradigm for training ICL
- The paper is supported by experimental analysis on simulated data related to linear regression to validate the theoretical results
- The paper investigated different scenario such as the case of noisy data

Weakness
- The paper derives the results for a single step of larger iterative algorithm
- The presented results are derived for the case of linear regression and it is unclear how it extends to non-linear problems.
- The experiments are conducted on toy datasets with normal distributions.
- The probing experiments are not very conclusive


**Summary Of The Paper:**

The paper investigates the capability of transformer-based in-context learners (ICL) to implicitly implement learning algorithms. The authors consider a study-case of linear regression models to validate their hypothesis that ICL encode context-specific parametric models: 1) ICL implement linear models by encoding gradient descent 2) ICL implements closed-form computation of regression parameters with Sherman–Morrison update. These results are summarized in Theorem 1 and 2 for a single-step update. The paper also investigates how the learned algorithm is implemented by investigating what intermediate quantities are being computed. The authors conducted probing experiments to assess if moment and weight vectors are computed for training linear regression ICL task.

**Summary Of The Review:**

The paper gives important insights on how ICL encode its learning mechanism. The theoretical results of the paper are derived in the case of linear regression. It is shown that ICL is cable of encoding gradient descent and regression parameters via closed-form. This contribution is useful for better understanding how transformers work in general and specifically in the case of In-Context-Learning.

---

> ### Author Response · Authors · 2022-11-15
> **Thank you!**
>
> Thank you for positive recommendations and valuable feedback!
>
> **What about multiple steps of SGD and Ridge Regression?**
>
> Given the complexity of constructive proofs, we resort to proving single-steps of these algorithms in the main body of the paper. However, in the original submission, we had provided Python codes that actually perform N-step SGD (Appendix A): https://icl1.s3.us-east-2.amazonaws.com/theory/sgd.py. In addition, we added a better discussion of N-step generalization in the updated Appendix A.
>
>
> **Why linear regression?**
>
> Thank you for this feedback. Linear regression presents an important test-bed for algorithmic understanding as it is extremely well understood, featuring closed form solutions for standard and Bayesian problem formulations, and many standard algorithms that we can compare ICL against. We hope that future work will extend this analysis to additional classes of learning problems.
>
> **More probing details?**
> Thank you for the suggestions, especially regarding controls for probing experiments. We have added results on probing a control model (trained to perform a non-ICL task) to the probing plots. Please see the updated Figure 4. We show that our probe’s accuracy is significantly higher than that of the control model. We have also fixed Eq. 18, which actually calculates input-independent attention weights to localize targets. Please see the updated Appendix E for further details.

---

### Official Review · Reviewer_p6X8 · 2022-10-25

**Confidence:** 4
**Correctness:** 3
**Technical Novelty And Significance:** 4
**Empirical Novelty And Significance:** 2
**Recommendation:** 8

**Clarity, Quality, Novelty And Reproducibility:**


The work in this paper is novel and of high quality. It is mostly clear but the probe technique leveraged in section 5 could use a lot more description to understand that set of experiments.

Minor issues and typos:

The plots and legend in figure 3 are hard to read because there are many methods plotted in the same graph with identical colors.

I believe there are some typos in the operator definitions of section 3.1. For instance:

1. in the "mov" operator's definition, should the matrix's first column read H_{:,:t-1} and last column read H_{:,t+1:}?  And should the middle column's top and bottom rows contain an i' and j' rather than an i and j, respectively?

2. for the "div" operator, the middle element is lacking proper subscripts and should be h_{i:j} / |h_{i'}|.

Section 4.3:

1. "m" should be capitalized to match "M" above

2. for weighted KNN, it looks like the kernel function is missing. I.e., K( |xi - xj|^2 )*yj instead of |xi - xj|^2 * yj.


**Strength And Weaknesses:**


Strengths:
This paper appears to be the first to investigate the neural mechansisms underpinning in-context learning for the very simple but non-trivial hypothesis class of linear regression models. The authors' finding that LLMs seem to mimic sensible and even Bayes-optimal learning algorithms is surprising and will surely motivate further research into understanding the power and limits of in-context learning.

Weaknesses:
As the authors admit, their findings are suggestive but the exact mechanisms of in-context learning for linear regression are still not understood.

Additionally, there were some experiments I was surprised the authors did not perform:
- examining how the dimensionality of the regression problem affects the learning algorithm derived for an LLM with a given capacity.
- determining whether the LLM derives Bayes optimal learning rule for other choices of p(w) and p(x). In particular, ridge regression is the correct choice for the p(w) used. But would the LLM mimic Lasso regression if p(w) was a Laplace prior over the regression coefficients?


**Summary Of The Paper:**


This paper presents an investigation into large-language models' (LLMs) abilities to learn algorithms for in-context linear regression given only sample problems consisting of inputs and linearly related outputs, but no information about the true (linear) hypothesis class underlying these relationships.

The authors prove that transformer LLMs are capable of representing and performing the matrix operations needed to support 2 commonly algorithms for linear regression -- gradient descent and regression via rank-1 updates.

And they also show that for a set of underdetermined regression problems generated by a particular (idealized) process, LLMs are capable of learning an algorithm for learning the minimum Bayes risk predictor (relative to that process). Further experiments are conducted to ellicit the effect of LLM model structure and capacity on the learning algorithms learned. The authors show that LLM capacity does influence which learning algorithms are derived, but how and to what extent is left unexplored.

Finally, the authors use a set of "probe" experiments to understand how in-context regression fitting works inside an LLM. The experiments suggest that the LLMs are computing fundamental quantities like the moment and weight vectors, but exactly how these are being computed (or if the hypothesis class is even truly linear) is not determined and is left for future work.


**Summary Of The Review:**

This paper is novel in that it provides the first examination of how in-context learning works for the simple but non-trivial learning problem of linear regression. Because in-context learning is a phenomenon that is just starting to be understood, and because this paper provides positive (though not conclusive) results, I advocate for acceptance to ICLR.

---

> ### Author Response · Authors · 2022-11-15
> **Thank you!**
>
> Thank you for the positive recommendations and valuable feedback!
>
> We have added additional experiments for the important questions asked in the review. We believe these experiments have strengthened the paper’s empirical conclusions—thanks for the recommendations. One set of final experiments, on Bayesian inference with alternative priors, are still in progress; we will update the paper again if these finish before the end of the review period.
>
> **How does problem dimensionality affect the implicit learning algorithm?**
>
> To answer this question, we experimented with dimension={1, 2, 4, 8, 12, 16, 20} and run an experiment sweep over:
>
> number of layers (L): {1, 2, 4, 8, 12, 16},
> hidden size (H): {16, 32, 64, 256, 512, 1024},
> number of heads (M): {1, 2, 4, 8},
> learning rate: {1e-4, 2.5e-4}.
>
> We present our results in the new Appendix H and Figure 8.
>
> For each feature that determines Transformer’s capacity (number of layers L, hidden dimension H, and number of heads M), we plot the minimum value (when the other features optimized) that makes model more close to OLS solution than a Ridge solution: $\operatorname{SPD}(\operatorname{OLS}, \operatorname{ICL}) < \operatorname{SPD}(\operatorname{Ridge}(\lambda=\epsilon), \operatorname{ICL})$. For $\epsilon=0.1$, we showed that the M=1 is enough for all dimensions, while L and H exhibit a step-function-like dependence on input size.
>
> In addition to these, we added $d=16$ results to Figure 3, and discuss hidden size requirements in Section 4.3
>
> **Is hypothesis class of ICL linear?**
> Thanks for this important question! We now discuss the linearity of the ICL in new Appendix F. We find that the learned function is non-linear for small numbers of examples but becomes linear by the time enough examples have been observed to fully determine the regression parameter.
>
> **More probing details?**
> Because of the page limit, we have added more details to the Appendix E. We additionally added control experiments to the probing plots which strengthens our findings. Please see the updated Figure 4. We fixed the Eq. 18 where we actually calculated input-independent attention weights to be able to localize the targets.
>
> **Legends and colors in plots**
> Thank you. We changed the color codes in the plots, the similar colors should correspond to the same class of algorithms.
>
> **Typos**
> Thank you! We fixed all the typos you mention. The kernel function in KNN was 1/distance. We can add exp(-distance) version in camera ready.

---

### Public Comment · ~Jian-Guang_Lou1 · 2022-11-07
**In-Context learning?**

The paper presents a very interesting topic, especially the probing results. I have a question about the experiment and the conclusion.

As mentioned by the authors, in-context learning focuses on the capability that a model can "map from sequences of (x, f(x)) pairs to accurate predictions f(x') on novel inputs x'."  Based on the definition the authors gave, to demonstrate the in-context learning capability, we need to make sure that the model learns a different linear function f(x') that is never seen in the training dataset based on the input examples in context.

The experiments in this paper may not be enough to prove this. In fact, for all linear function examples, the model only needs to learn a single function to handle all cases rather than different linear functions for different cases. For example, given a set of [x_i, y_i], and x_n, we want to predict the value of y_n. Because (y_i-y_0)/(x_i-x_0)=(y_n-y_0)/(x_n-x_0) is always correct for all linear functions, therefore, the model only needs to learn a single function (e.g., y_n = y_0+(x_n-x_0)*(y_i-y_0)/(x_i-x_0) for no-noise experiments) to predict y_n rather than different f(x') for different input context.

In the experiments, each in-context dataset is a sequence of 40 (x, y) pairs. All training samples and testing samples have the identical length, which increases the possibility of that the model only learns a single model for all cases including training and testing data.

I will suggest to conduct experiments on varying length of in-context dataset to make the experiment more strong. Especially, the length of in-context dataset in the test should be different from that in the training. If the experiments still give positive results, the claim of this paper may be more convincing.

---

> ### Author Response · Authors · 2022-11-09
> **Thanks for the constructive feedbacks**
>
> Thank you for your interest and comments on the paper!
>
>
> **Do these regression problems have a trivial solution?**
>
> We believe there is a misunderstanding: while the solution described in the comment applies to 1-dimensional regression problems, all problems that we consider in our experiments are multi-dimensional (e.g. $x \in R^{8}, w \in R^{8 \times 1}$). A $d=8$ linear regression problem is underdetermined with 2 data points, and there are many different regression parameters that exactly fit those two points. The multidimensional linear regression problem and its solutions are discussed in Section 2.3 and 3.3. Moreover, we show that ICL can solve the *noisy* linear regression in the Bayesian sense (Figure 3). Thank you for pointing out that this is unclear; we will update the above-mentioned sections to clarify further.
>
> **Is the model over-fit to a specific training set size?:**
>
> Great question! The statement “all training samples and testing samples have identical length” is incorrect. All experiments in this paper use a Transformer *decoder* (in which attentions are causally masked) with *auto-regressive* language modeling objective. When we say that we use sequences with $n=40$ in training, we mean that the loss is taken for all $\hat{y}_n$ given $n-1$ exemplars up to $n=40$. In other words, $n=40$ is a maximum sequence length; because of the autoregressive objective, models are effectively trained (and evaluated) on sets with size ranging from 1 to 40. Figure 1 shows model behavior for varying in-context set sizes $n$ (x-axis). We will add additional text clarifying this point in Section 4.2. For more details on this, please refer to the very recent work that uses the same experimental setup: Garg et al. 2022 (https://arxiv.org/abs/2208.01066).
>
> Best regards,
> Paper Authors

---

> > ### Public Comment · ~Jian-Guang_Lou1 · 2022-11-15
> > **Thanks for reply!**
> >
> > Thanks a lot for the clarification!
> >
> > I really appreciate the insightful observations in the paper, such as that the model can solve the noisy linear regression in the Bayesian sense. Even without "in-context" claim, the paper is still very interesting. What I am curious is about whether this is really "In-Context" or not. It depends on the definition of "in-context learning": what is the key difference from "in-context learning" to the traditional one?  The previous comments may not be very clear, but don't get me wrong. The goal of my 1-dimensional example is to give a very special case to illustrate the idea that the model only needs to learn the same function (for a length of context pairs) that works for both training and testing samples. Although 1-dimensional example is a very special case, but this function invariance is generic to different dimensions.
> >
> > In terms of pair length, thanks for the further clarification. I may need to clarify my point in the comment. In section 4.2, the paper mentions "We follow the training guidelines in Garg et al. (2022), and trained our models for 500, 000 iterations, where each in-context dataset is a sequence of 40 (x, y) pairs.". It mean the length of in-context dataset of both training and test samples are 40. Therefore, all the length values (i.e., 1-40) in test are seen in the training. As mentioned above, it is possible that the model only learns a single function for each length that can work for both training and testing samples. Therefore, I am curious about the performance on a test case with a context of different length e.g., 41 pairs.
> >
> > This is all about my curiosity about what the model really learns from the data. Anyway, the observation that the model works like a minimum-Bayes-risk predictor in this paper is very interesting.

---

> > > ### Author Response · Authors · 2022-11-19
> > > **thank you for the clarification**
> > >
> > > Thanks for the clarification and opening this interesting discussion up!
> > >
> > > **Is the model trained in this paper learning to perform a single function for each in-context datasets? What kinds of functions actually count as in-context learning?**
> > >
> > > Consider a learning problem with fixed-dimensional inputs and a dataset with a variable number of examples  $n$. A deterministic learning algorithm is a a **higher-order function** that takes a  dataset $[x_0, y_0, .., x_n, y_n]$ and outputs another function $f$ which is the predictor.
> > >
> > > In this sense, any learner is "just a single function" that happens to accept variable-length lists as input. The complexity of this function is dependent on the learning problem that it needs to solve.
> > >
> > > In the case of general linear regression, at optimality, we expect the model to (implicitly) implement a single higher-order function, $\mathcal{A}(D)=\hat{w}=(X^TX)^{-1}X^Ty$, to calculate $\hat{w}$ and then predict $y$s with this $\hat{w}$.
> > >
> > > In fact, in the noiseless one-dimensional case, optimal prediction rule corresponds exactly to the function in your parent comment: $y_n = y_0+(x_n-x_0)*(y_i-y_0)/(x_i-x_0)$ — in 1D, *any* a successful in-context learner must be behaviorally equivalent to this function (but might perform a different sequence of arithmetic operations to compute it).
> > >
> > > In summary, what we and previous work call "in-context learning" is being able to (implicitly) infer different predictor functions (i.e. different $w$s) in-context (without any parameter updates)
> > >
> > > [Note that $f$ (or $w$) is constant within each in-context dataset, but *different* across in-context datasets — we evaluate our model for such different $f$s (or $w$s) in Figure-1 and present the average results over such various $w$s.]
> > >
> > > **3) Does the ICL trained in this paper generalizes to $n>40$ in-context datasets?**
> > >
> > > You are referring to general phenomena with neural sequence models: “length generalization” (Lake & Baroni 2018, Newman et al. 2020, Csordás et ak 2021, Anil et al. 2022). In Transformers, using absolute embeddings causes this problem, and there are various tricks to alleviate this problem (e.g. Press et al. 2021). Because we use absolute learned positional embeddings, we do not expect length generalization in our ICLs. We agree this is an important feature direction — understanding out-of-distribution generalization capabilities of Transformers under ICL.
> > >
> > > **4) If an ICL doesn’t length-generalize, can we say that it is doing in-context learning?**
> > >
> > > Length generalization and in-context learning ability is orthogonal to each other. We can prove that there is a Transformer that performs SGD on previous examples (with batch size=1) and predicts according to SGD updated predictor, but behaves completely differently after seeing $n$ examples.
> > >
> > > For a preliminary, please see our constructions in Appendix A (Theorem-1) for SGD (see the supplementary code for generalization to n-step SGD) and the details of how we constructed each time-step’s attention in C.4.1 (Equations 29-30).
> > >
> > > In this SGD construction, we can set positional embeddings $p_{i>2n}^{(l)}=q_{i>2n}^{(l)}=e_i$ for all $l$; meaning that every position after $2*n$ will attend to only itself. In other words, the model will not look back after $n$th exemplar, and SGD will not operate correctly after this point.
> > >
> > > This model perfectly performs SGD, so does in-context learning up to $n$th exemplar, and does not generalize after $n$th exemplar.

---

### Author Response · Authors · 2022-11-19
**Rebuttal Summary**

We thank all the reviewers for their detailed feedback and positive recommendation for our work. We are glad that **p6X8** found our work “novel and high quality”, and **NUsG** suggested a *highlight* in the conference.

We tried to answer all the requests from the reviewers. We run the suggested experiments – as a result we have a **better understanding of the underlying ICL predictor**. Important highlights from this rebuttal period were:

- We added **dimensionality analysis** and displayed the empirical scaling (e.g. how many hidden sizes are required for a given dimensionality) for the Transformer to get close to the OLS solution (Figure 8, Appendix H).

- We strengthen our probing results by adding a **control probe**. We can observe that ICL is encoding expected parameters much better than this control model (Figure 4).

- We found that ICL is **getting gradually linear** in the undetermined regime (Appendix F, Figure 6)

- We generalized our construction: now, we do not need to assume GeLU non-linearity (Appendix G)

- We improved the Figure 2 (Bayesian LS) by changing it to a heatmap (Figure 2)

---

### Decision · Program_Chairs · 2023-01-20

**Decision:**

Accept: notable-top-5%

**Justification For Why Not Higher Score:**

N/A

**Justification For Why Not Lower Score:**

I believe this can be of interest to many in the community trying to understand what Transformers are learning.

**Metareview: Summary, Strengths And Weaknesses:**

This work investigates why transformers can be trained to be in-context learners (ICL) of linear functions. Specifically, the authors provide different perspectives to support the hypothesis that transformers can learn standard learning algorithms and low-lever algorithmic operations in this way. E.g., a bottom-up construction implying that optimization techniques can be "embedded" in their forward pass or testing how similar the outputs of transformers are to well-known algorithms.

All reviewers agreed that the paper tackles an important question, and while not providing a definite answer, brings interesting perspectives that can further boost research in the direction of understanding what transformers are learning. Some concerns about the presentation or the assumptions regarding the theoretical analysis (e.g., using GeLU as non-linearities) were raised in the reviews.

Authors were responsive during rebuttal, managed to address all the concerns and uploaded a stronger version of the paper including improved presentation, a dimensionality analysis, controlled experiments and a more generalized analysis not requiring to assume GeLU activations as requested by one reviewer.

The paper is therefore accepted.

**Note From Pc:**

if the above contains the word "oral" or "spotlight" please see: "oral" presentation means -> notable-top-5% and "spotlight" means -> notable-top-25%. As stated in our emails, we are disassociating presentation type from AC recommendations

**Summary Of Ac-Reviewer Meeting:**

N/A